# INTERNSPATIAL: A COMPREHENSIVE DATASET FOR SPATIAL REASONING IN VISION-LANGUAGE MODELS

**Nianchen Deng**[1*]  **Lixin Gu**[1*]  **Shenglong Ye**[1*]  **Yinan He**[1]  **Zhe Chen**[2,1]
**Songze Li**[3,1]  **Haomin Wang**[3,1]  **Jinhui Yin**[2,1]  **Qi Wei**[2,1]  **Tianshuo Yang**[1]  **Min Dou**[1]
**Tong He**[1]  **Wenqi Shao**[1]  **Kaipeng Zhang**[1]  **Yi Wang**[1]  **Botian Shi**[1]  **Yanting Zhang**[4]
**Jifeng Dai**[5,1]  **Yu Qiao**[1]  **Wenhai Wang**[6,1†]  **Hongjie Zhang**[1†]

[1] Shanghai AI Laboratory  [2] Nanjing University  [3] Shanghai Jiao Tong University
[4] Donghua University  [5] Tsinghua University  [6] The Chinese University of Hong Kong
{dengnianchen,gulixin,zhanghongjie}@pjlab.org.cn

## ABSTRACT

Recent benchmarks and datasets have been proposed to improve spatial reasoning in vision-language models (VLMs), yet existing open resources remain constrained by limited scale, narrow visual diversity, and restricted instruction expressiveness. To address these gaps, we present InternSpatial—the largest open-source dataset for spatial reasoning in VLMs—alongside InternSpatial-Bench, a comprehensive evaluation benchmark designed to assess spatial understanding across diverse instruction formats. InternSpatial contains 12 million question-answer(QA) pairs covering both single-view and multi-view scenarios, sourced from varied visual environments and supporting 19 distinct instruction formats that mirror real-world query patterns. InternSpatial-Bench aims to single-view assessment and also extends multi-view reasoning through a novel rotation estimation task. Experimental validation demonstrates that models trained on InternSpatial achieve substantial performance improvement of 12.1% on InternSpatial-Bench and 10.7% on VSI-Bench, while preserving competitive performance on general-purpose benchmarks. We expect these resources can advance the development of spatially-capable VLMs for practical applications in robotics and embodied AI systems. Our codes and datasets are publicly available at https://github.com/dengnianchen/intern-spatial.

## 1 INTRODUCTION

Vision-language models (VLMs) have achieved remarkable progress across multimodal tasks such as visual question-answering (VQA), image captioning, and grounding. However, they continue to struggle with spatial reasoning in both single-view settings (*e.g.* identifying object position or size from a static image) and multi-view scenarios (*e.g.* estimating distances or tracking appearance order across video frames). Enhancing spatial reasoning capabilities is crucial for real-world applications such as robotics, autonomous navigation, and augmented reality.

Recent efforts have introduced spatially-relevant VQA datasets and benchmarks(Cai et al., 2025; Cheng et al., 2024; Chen et al., 2024a; Yang et al., 2024). While these works have advanced the field, several limitations still persist: (1) *Limited scene diversity*: existing datasets are predominantly drawn from narrow sources, primarily indoor or outdoor scenes; (2) *Restricted instruction formats*: SpatialVLM (Chen et al., 2024a) and SpatialQA (Cai et al., 2025) rely exclusively on natural language, and OSD (Cheng et al., 2024) uses region masks, which fail to reflect the diversity of instruction formats required for practical tasks; (3) *Narrow training scope*: existing datasets mainly focus on single-view settings and basic spatial concepts such as object position or existence, lacking multi-view supervision for spatial relationships across viewpoints.

---

[*] Equal Contribution.
[†] Corresponding Author.

Table 1: Comparison of our InternSpatial with existing spatial reasoning datasets. W: in-the-wild, I: indoor, D: drive, E: embodied, O: object-centric

| Dataset | # of QA | Scenario | Open-source | View Type | Instruction format |
|---------|---------|----------|-------------|-----------|--------------------|
| SpatialVLM (Chen et al., 2024a) | 2B | W | ✗ | Single-view | Single-format |
| SpatialQA (Cai et al., 2025) | 0.9M | W,E | ✓ | Single-view | Single-format |
| OSD (Cheng et al., 2024) | 8.7M | W | ✓ | Single-view | Single-format |
| InternSpatial | 12M | W,I,D,E,O | ✓ | Single-view, Multi-view | Multiple-format |

To address these challenges, we introduce InternSpatial– the largest open-source spatial reasoning dataset, and a corresponding benchmark *InternSpatial-Bench*, specifically designed to enhance spatial reasoning in VLMs. InternSpatial comprises 9.5M single-view and 2.5M multi-view QA pairs, sourced from diverse visual environments including in-the-wild scenes (Lin et al., 2014; Wang et al., 2024c; Krishna et al., 2017), structured indoor spaces (Wald et al., 2019; Dai et al., 2017; Mao et al., 2022), urban streetscapes (Cordts et al., 2016), object-centric scenes (Deitke et al., 2022), and embodied navigation contexts (Anderson et al., 2018). To enrich instruction formats, we incorporate a diverse set of query representations: masks, bounding boxes, numerical indicators embedded in images, as well as textual coordinate-based references and spatial cues. Overall, the dataset supports 19 distinct instruction formats, enabling broader coverage of spatial reasoning queries. We further introduce a novel cross-view rotation estimation task, with 2.46M newly collected training QA pairs, addressing a gap in existing datasets. To facilitate evaluation, we construct InternSpatial-Bench with 6,008 QA pairs, serving as a comprehensive diagnostic benchmark for single- and multi-view spatial reasoning tasks. As shown in Table 1, InternSpatial significantly expands scene coverage, instruction diversity, and multi-view supervision compared to existing benchmarks.

Our contributions are threefold: (1) The largest open-source spatial reasoning dataset InternSpatial for VLMs, designed for supervised fine-tuning with diverse scenes and 19 instruction formats; (2) A comprehensive evaluation benchmark InternSpatial-Bench for both single-view and multi-view tasks, including the novel rotation estimation task; (3) Extensive experiments demonstrating the effectiveness of InternSpatial, achieving 12.1% improvement on InternSpatial-Bench and 10.7% on VSI-Bench while maintaining general multimodal performance.

## 2 RELATED WORK

### 2.1 SPATIAL REASONING VIA VISION LANGUAGE MODELS

Recent advances in large language models (LLMs) (Brown et al., 2020; Achiam et al., 2023; Touvron et al., 2023) and vision-language models (VLMs) (Zhu et al., 2022; Li et al., 2023a; Zhu et al., 2023a; Wang et al., 2023; Liu et al., 2023; Li et al., 2023b; Wang et al., 2024c; Chen et al., 2024c) have demonstrated remarkable capabilities. However, growing evidence indicates that VLMs still struggle with spatial reasoning tasks (Cai et al., 2025; Chen et al., 2024a; Cheng et al., 2024; Yang et al., 2024). To address these issues, recent works explicitly inject spatial awareness. Ferret-v2 Zhang et al. (2024a) improves fine-grained regional referring and grounding with any-resolution visual encoding. LocVLM (Ranasinghe et al., 2024) enhances spatial reasoning via coordinate-based instruction tuning. Shikra (Chen et al., 2023) represents spatial coordinates purely in natural language for referential dialogue. Beyond 2D localization, several approaches incorporate additional supervision signals such as dense 3D representations (Hong et al., 2023b;a), image space masks (Cheng et al., 2024), and pixel-wise depth information (Cai et al., 2025). Despite these efforts, current methods still fall short of comprehensive, end-to-end spatial reasoning across diverse single-view and multi-view scenarios, which motivates the design of our InternSpatial dataset.

### 2.2 SPATIAL REASONING DATASETS

To evaluate and enhance the spatial reasoning capabilities of VLMs, several datasets and benchmarks have been proposed. SpatialEval (Wang et al., 2024a) targets 2D spatial reasoning across relation understanding, navigation, and counting tasks. On the other hand, TopViewRS (Li et al., 2024c) explores spatial reasoning from a top-down perspective, emphasizing the need to enhance VLM performance in top-view settings. For 3D spatial understanding from images, several datasets have been introduced (Cheng et al., 2024; Cai et al., 2025; Li et al., 2024d), primarily tailored to specific

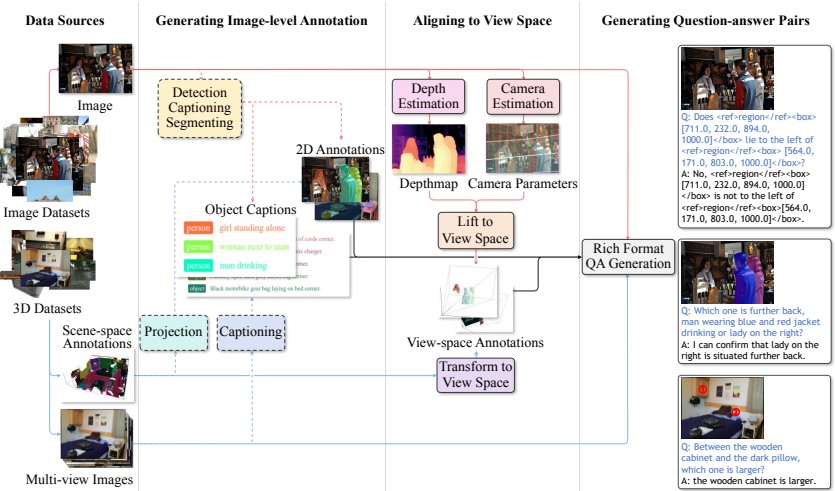

Figure 1: Generation pipeline for InternSpatial. The optional flows (represented by dashed lines and boxes) are only performed when the relevant annotations does not exist in the data source.

models and require additional inputs like segmentation masks or depth maps. SpatialVLM (Chen et al., 2024a) proposed a large-scale 3D spatial VQA dataset constructed by an automatic data generation framework using Internet images (Chen et al., 2024a), demonstrating that appropriate training data can help VLMs infer spatial relationships without auxiliary inputs. Unfortunately, this dataset remains publicly unavailable. Spatial reasoning over image sequences or videos presents additional challenges. To assess such capabilities, the VSI benchmark (Yang et al., 2024) was proposed, evaluating a range of open-source and proprietary VLMs. Results show that current models still struggle with multi-frame spatial reasoning tasks. Our work addresses these limitations by introducing a dataset integrating both single-view and multi-view tasks, significantly enhancing VLM spatial reasoning across diverse contexts and highlighting their potential for deeper spatial understanding.

# 3 DATASET

## 3.1 DATA ENGINE FOR INTERNSPATIAL

We construct InternSpatial, a large-scale dataset comprising nearly 12 million Question-Answer(QA) pairs, to enable VLMs to perform 3D spatial reasoning through supervised fine-tuning. InternSpatial aggregates data from a wide range of sources, including in-the-wild scenes (Lin et al., 2014; Wang et al., 2024c; Krishna et al., 2017), structured indoor spaces (Wald et al., 2019; Dai et al., 2017; Mao et al., 2022), urban streetscapes (Cordts et al., 2016), object-centric scenes (Deitke et al., 2022), and embodied navigation contexts (Anderson et al., 2018).

To handle the heterogeneity of source data and support large-scale QA generation, we develop a fully automated and modular data engine that consolidates intermediate annotation extraction and QA synthesis into a unified pipeline applicable across diverse data sources. As illustrated in Figure 1, the pipeline begins by generating necessary annotations at the image level, followed by transforming the annotations into a canonical view space. Finally, QA pairs are constructed using a template-based approach that supports a wide variety of task types and instruction formats.

**Generating Image-level Annotation.** To generate 3D spatial reasoning QAs grounded in objects, we first obtain the necessary image-level annotations, including 2D bounding boxes, region descriptions, segmentation masks, etc. For image datasets that already provide such annotations, we directly utilize the existing labels. When annotations are missing, we employ pretrained models to generate them automatically. Specifically, we use open-source VLMs to extract object-level 2D boxes and associated textual descriptions, and apply the SAM2 model (Ravi et al., 2024) to generate segmentation masks within these boxes. These masks are subsequently lifted into 3D space to facilitate the construction of 3D bounding boxes. The prompts we used in this step can be found in

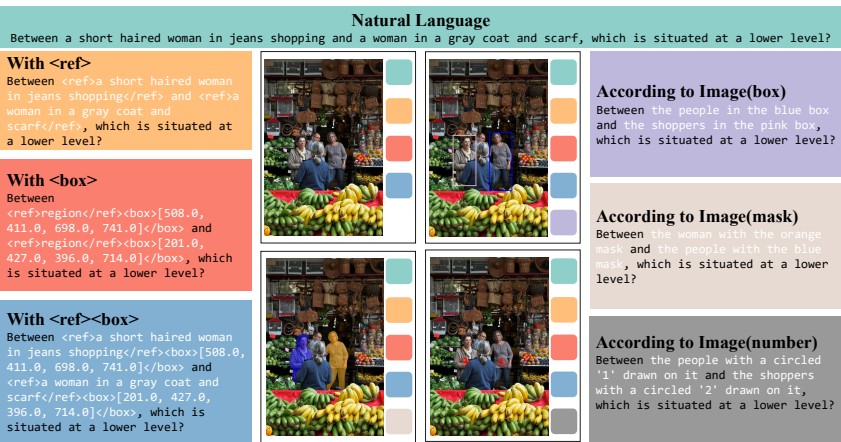

Figure 2: Examples of diverse instruction formats in text and image. The four images illustrate different visual formats: original (top-left), bounding boxes (top-right), segmentation masks (bottom-left), and numbered regions (bottom-right). Surrounding the images are seven corresponding text instruction formats. The color blocks beside each image indicate whether the corresponding image-text pair is included in InternSpatial and InternSpatial-Bench. Best viewed in color.

Appendix C. In the case of 3D datasets, which typically include global 3D annotations and per-view camera parameters, we project the 3D information onto the image plane to obtain the corresponding 2D annotations. Although this projection is not strictly required for generating QAs, as the underlying 3D annotations are already available, it is necessary for supporting visual reference forms in prompts, such as bounding boxes and segmentation masks. In order to reduce potential ambiguity in questions, We further apply several filtering strategies. For in-the-wild images, we filtered out objects without clear boundaries by object category, such as the sky and grass. For indoor scenes with 3D annotations, we detected the occlusions and objects beyond the image boundary through the projection of 3D models and bounding boxes on the image plane, and excluded QA pairs that were ambiguous due to these situations.

**Aligning to View Space.** To determine spatial relationships between objects, it is essential to obtain their positions and dimensions within a well-defined 3D coordinate system. We adopt a canonical view space as the reference frame, defined as a 3D Cartesian coordinate system centered at the camera's optical center. In this space, the y-axis aligns with the viewing direction, and the z-axis is perpendicular to the scene's horizontal plane, pointing upward. For 3D datasets, which provide global annotations and per-view camera parameters, transforming annotations into the canonical view space is straightforward. In contrast, image-only datasets contain only 2D visual information, requiring estimation of both camera parameters and depth maps. To address this, we follow the pipeline of SpatialRGPT (Cheng et al., 2024), leveraging WildCamera (Zhu et al., 2023b) for intrinsic parameter estimation, PerspectiveFields (Jin et al., 2023) for extrinsic parameter inference, and Metric3Dv2 (Hu et al., 2024) to predict dense depth maps. By combining the outputs of these models, we lift 2D annotations into the canonical 3D space, enabling accurate reasoning over object-level spatial relationships.

**Template-based QA Generation.** While prompting a large language model (LLM) to generate QA pairs directly for each image can produce diverse instructions, this approach is prohibitively expensive at scale in terms of computation and time. Instead, we adopt a template-based generation strategy that avoids invoking the LLM during QA construction. This approach not only improves efficiency but also facilitates flexible expansion to multiple prompt styles, such as object references via bounding boxes or segmentation masks. To ensure sufficient instruction diversity, we first prompt an LLM to generate several QA templates for each task type and answer format. These templates contain placeholders for object references and other variable content. During generation, we randomly select a subset of tasks and object instances (or pairs) for each image, derive the corresponding answers using the previously constructed annotations, and instantiate the templates accordingly. We then filter out low-quality QA pairs, such as those involving ambiguous spatial relationships caused

by occlusion, and balance the number of positive and negative examples to produce a well-structured dataset. We generate templates for 4 single-view tasks, covering the position/size relationship of two objects, as well as relationship-constrained count and existence tasks. The list of templates are shown in Appendix B.

**Extending Instruction Formats.**     To enhance dataset diversity and better reflect real-world usage scenarios, we extend each QA pair into multiple instruction formats. Specifically, we generate up to five textual formats and up to four image formats per QA pair. The image formats include: (1) the original image, (2) the image annotated with bounding boxes, (3) the image with segmentation masks, and (4) the image annotated with numbers over key objects. The textual formats include: (1) natural language descriptions, (2) text with `<ref>{caption}</ref>` (3) text with `<ref>region</ref><box>{bbox}</box>` (4) text with `<ref>{caption}</ref><box>{bbox}</box>` and (5) text automatically generated based on image content. Representative examples of these visual and textual formats are shown in Figure 2. As a result, each QA pair can produce up to 19 training samples, from which only suitable ones are retained. Additionally, certain prompt types, such as images with numbers on key objects, may not directly indicate the correct object. Therefore, in these cases, we utilize the position information from the segmentation mask to correctly identify and reference the target object. During training, we uniformly sample across all instruction formats.

**Generating Multi-view QA Pairs.**     To develop a comprehensive multi-view dataset for spatial understanding, we systematically collected and integrated multi-view data derived from the training splits of the ScanNet (Dai et al., 2017), MultiScan (Mao et al., 2022), R2R (Anderson et al., 2018), and Objaverse (Deitke et al., 2022), subsequently formulating temporally-agnostic training samples that encapsulate inter-object relational attributes such as relative properties, scale variations, and spatial distances, and cross-view relationships of objects such as rotation. Scene-level geometric priors were established by estimating room dimensions via the Alpha Shape algorithm (Akkiraju et al., 1995) applied to the point clouds, with the room centroid defined as the geometric center of the minimal axis-aligned bounding box enclosing the scene. We meticulously cataloged instance counts for each object semantic category. For unambiguous objects within the point clouds exhibiting a principal dimension exceeding 15cm, annotations were standardized to the *OrientedBoundingBox* format using Open3D (Zhou et al., 2018). For remaining objects or those with initial ambiguities, we leveraged existing annotations to reduce the risk of shortcut learning by language models. Plausible alternative options were constructed by extracting distractors from other items within the dataset, thereby forming a corresponding multiple-choice question training set.

**Human validation.**     Due to the huge number of QAs in our dataset, it's almost impossible to check all generated QAs by human. Instead, we conducted a manual verification on a randomly sampled subset with 500 items, including both final QAs and intermediate steps. With this validation, we ensure the accuracy of QAs in our dataset was over 95%.

## 3.2 INTERNSPATIAL-BENCH

To evaluate the performance of VLMs on 3D spatial reasoning tasks, particularly under diverse instruction formats, we propose InternSpatial-Bench, a novel multi-task benchmark that features a broad range of input types. Existing benchmarks such as SpatialRGPT-Bench (Cheng et al., 2024) and SpatialBench (Cai et al., 2025) present several limitations. First, the question formats are overly simplistic and do not reflect real-world application scenarios. Second, these benchmarks are tailored to specialized models and require auxiliary inputs such as region masks or depth maps. As a result, many tasks are incompatible with general-purpose VLMs that operate solely on images and text. Furthermore, SpatialBench suffers from a limited number of QA pairs, reducing its effectiveness as a comprehensive evaluation suite.

InternSpatial-Bench expands and refines both SpatialRGPT-Bench and SpatialBench to overcome these limitations. Specifically, we enrich instruction formats and introduce 3,000 carefully curated QA pairs, resulting in a total of 5,300 high-quality examples that span diverse task types and input modalities. Certain tasks from the original benchmarks, such as reachability prediction and quantitative estimation of spatial extent, are excluded because they are unsuitable for general-purpose VLMs when only a single-view image is provided. In the absence of additional information, such as

depth or camera parameters, these tasks become severely under-constrained and often ambiguous, even for human annotators.

**Refining and Expanding SpatialRGPT-Bench and SpatialBench.** Since SpatialRGPT-Bench (Cheng et al., 2024) already provides a sufficient number of QA pairs, our focus is on expanding the diversity of question formats rather than increasing the dataset size. Specifically, we augment the instruction styles of the original questions that do not involve numerical reasoning, following the format extension strategy described in subsection 3.1. However, to avoid ambiguity caused by duplicate object labels, we exclude formats that rely on natural language references or textual content containing `<ref>caption</ref>`. For each selected question, we randomly sample three different formats and leverage both object mask and bounding box annotations to construct the final benchmark entries. SpatialBench (Cai et al., 2025) contains QA pairs exclusively in natural language form. To diversify its instruction formats, we first manually extract reference phrases corresponding to the mentioned objects and convert the questions into templates with placeholders. Next, we prompt the VLM to ground the objects based on these phrases and apply SAM2 to segment the corresponding regions. Using the resulting question templates, along with object bounding boxes and masks, we apply the format extension method described in subsection 3.1 to generate diverse instruction variants for each QA. Finally, all generated QA pairs are manually verified to ensure quality, with erroneous answers corrected and ambiguous or ill-formed questions removed

**Extending the Benchmark with Curated QA Pairs.** Unlike the large-scale training dataset, the benchmark is relatively small but demands higher annotation quality. To this end, we implement a dedicated pipeline for generating high-quality QAs used in the benchmark. This pipeline operates without relying on any pre-annotated information, making it applicable to any image-only data source. To encourage diversity and expressiveness in question formulation, we prompt the VLM to generate questions directly. Finally, we introduce a manual verification step to review all automatically constructed questions and answers, ensuring the overall quality and correctness of the benchmark data. Details of the construction process are provided in Appendix D.

## 3.3 DATASET STATISTICS

**Statistics of InternSpatial.** Our proposed dataset, InternSpatial, encompasses a diverse set of tasks and instruction formats to comprehensively enhance spatial reasoning capabilities. It consists of a total of 12,035,415 QA pairs, covering both single-view and multi-view spatial reasoning tasks. Specifically, the single-view tasks include *Position Comparison*, *Size Comparison*, *Existence Estimation*, and *Object Counting*, while the multi-view tasks include *Rotation Estimation*, *Object Counting*, *Room Size Estimation*, *Object Size Estimation*, *Route Planning*, and *Appearance Order*. Detailed task descriptions and corresponding statistics are provided in Appendix A, and visual examples are shown in Appendix F. In addition, InternSpatial incorporates images from various sources to enhance the robustness of the model. As illustrated in Figure 3, the dataset includes COCO (Lin et al., 2014), AS-1B (Wang et al., 2024c), and Visual Genome (VG) (Krishna et al., 2017) for in-the-wild imagery; 3RScan (Wald et al., 2019), ScanNet (Dai et al., 2017), and MultiScan (Mao et al., 2022) for indoor scenes; Cityscapes (Cordts et al., 2016) for street scenes; Objaverse (Deitke et al., 2022) for single-object scenarios; and R2R (Anderson et al., 2018) for embodied navigation tasks. Moreover, InternSpatial emphasizes diversity in instruction formats. As shown in Figure 3, the number of samples across different formats is carefully balanced to avoid bias and ensure uniform coverage during training. In summary, InternSpatial provides a large-scale, diverse resource spanning task types, visual domains, and instruction formats, making it well-suited for training VLMs to handle real-world spatial reasoning tasks effectively.

**Statistics of InternSpatial-Bench** Following Spatial-Bench and Spatial-RGPT, our proposed benchmark, InternSpatial-Bench, includes five tasks—*Position Estimation*, *Size Estimation*, *Rotation Estimation*, *Existence Estimation*, and *Object Counting*—designed to systematically evaluate the spatial reasoning capabilities of VLMs. In total, InternSpatial-Bench consists of 6,008 QA pairs. Detailed task statistics are provided in Appendix A, and visual examples are shown in Appendix G. To ensure robustness and diversity, InternSpatial-Bench incorporates images from a broad range of domains. As shown in Fig 4, in addition to the sources used in Spatial-Bench and Spatial-RGPT, we include samples from the test sets of COCO, Flickr30K, Objaverse, ScanNet, and Cityscapes. This

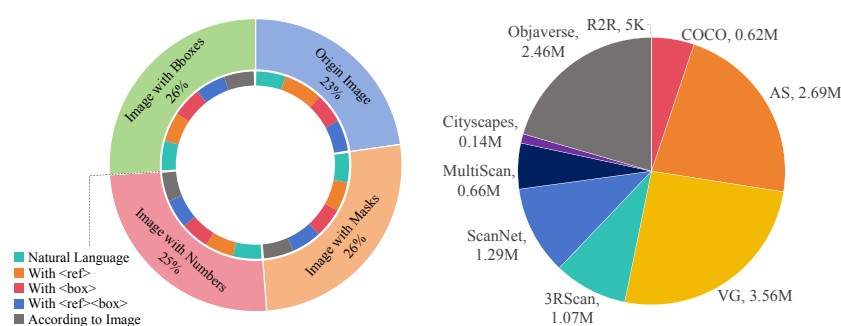

Figure 3: Distribution of instruction formats (**Left**) and data sources (**Right**) in InternSpatial.

diverse image collection spans a wide range of real-world contexts, from indoor and outdoor environments to single-object scenarios and in-the-wild imagery. We apply the same instruction format expansion strategy as used in InternSpatial, with one exception: for the Rotation Estimation task, since each image contains only a single object, we only use the original image format and natural language instructions. Consequently, these formats have a higher proportion in this task compared to others. By combining diversity in task types, visual domains, and instruction formats, InternSpatial-Bench offers a comprehensive and realistic benchmark for evaluating the spatial reasoning abilities of VLMs across a wide range of practical scenarios.

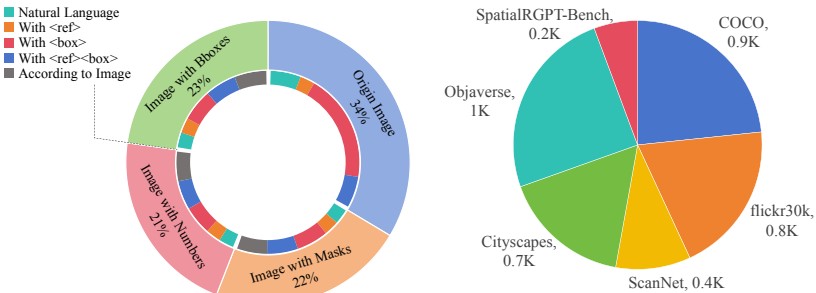

Figure 4: Distribution of instruction formats (**Left**) and data sources (**Right**) in InternSpatial-Bench.

## 4 EXPERIMENTS

We begin in Section 4.1 by introducing the baseline model and outlining the evaluation benchmarks used in our experiments. Section 4.2 then presents results on InternSpatial-Bench to assess the spatial reasoning capabilities of vision-language models. Section 4.3 reports performance on VSI-Bench (Yang et al., 2024), which further evaluates the models' multi-view spatial reasoning abilities. In Section 4.4, we conduct an ablation study to analyze the impact of different instruction formats on model performance. Finally, Section 4.5 evaluates whether training with InternSpatial affects general reasoning ability by benchmarking against a suite of standard vision-language tasks.

### 4.1 EXPERIMENT SETUP

**Baseline.** We construct our baselines based on InternVL2.5-8B (Chen et al., 2024c), a representative traditional VLM. Following the training settings of InternVL2.5, we fine-tune our models from InternVL2.5-8B using a downsampled version of the general datasets employed in InternVL2.5, along with InternSpatial. For generality, we also utilize InternVL2.5-1B and Qwen2.5-VL-8B as additional baselines to demonstrate the transferability of our training strategy. We refer to the models fine-tuned on InternSpatial as InternVL-Spatial-8B (for InternVL2.5-8B), InternVL-Spatial-1B (for InternVL2.5-1B), and Qwen-Spatial-8B (for Qwen2.5-VL-8B (Bai et al., 2025)). Detailed training configurations are provided in Appendix E.

**Evaluation.** We evaluate the models trained on InternSpatial using three types of benchmarks: our proposed InternSpatial-Bench, the multi-view spatial reasoning benchmark VSI-Bench (Yang

et al., 2024), and several general-purpose benchmarks, including MathVision (Wang et al., 2024b), OCRBench (Liu et al., 2024), TextVQA (Singh et al., 2019), ChartQA (Masry et al., 2022), and MM-Star (Chen et al., 2024b). For InternSpatial-Bench, we follow the evaluation protocols of Spatial-Bench (Cai et al., 2025) and Spatial-RGPT (Cheng et al., 2024), reporting relative error for counting tasks, accuracy for multiple-choice questions, and GPT-4o-assigned (OpenAI, 2025) scores for quiz-style questions. For VSI-Bench, we adopt the official evaluation protocol, with the only modification being the use of 32 sampled frames per video during testing. For general benchmarks, we follow the evaluation procedures provided by OpenCompass (Contributors, 2023).

## 4.2 EVALUATION ON INTERNSPATIAL-BENCH

Table 2: Results on InternSpatial-Bench. **Bold** indicates the best performance among all models, while underline denotes the second-best performance.

| Model | Position Comparison | Size Comparison | Rotation Estimation | Object Counting | Existence Estimation | Average |
|---|---|---|---|---|---|---|
| Humnn Level | 99.7 | 97.7 | 100.0 | 98.9 | 100.0 | 99.3 |
| GPT-4o-2024-11-20 (OpenAI, 2025) | 71.2 | 71.5 | 26.7 | 63.5 | 74.9 | 61.6 |
| Claude-3.7-Sonnet-20250219 (Anthropic, 2024) | 73.2 | 72.3 | 25.9 | 59.2 | 70.5 | 60.2 |
| Gemini-2.5-Flash(Comanici et al., 2025) | 64.5 | 67.3 | 30.2 | 67.0 | 67.3 | 59.3 |
| Llama-4-Scout(Meta Platforms, 2025) | 42.2 | 45.0 | 20.8 | 44.0 | 25.7 | 35.5 |
| Qwen2.5-VL-72B (Bai et al., 2025) | 54.6 | 55.3 | 30.6 | 60.5 | 63.3 | 52.9 |
| Pixtral-12B (Agrawal et al., 2024) | 65.6 | 62.9 | 5.8 | 52.5 | 78.3 | 53.0 |
| LLaVA-OneVision-72B(Li et al., 2024b) | 77.8 | 77.0 | 25.8 | 64.5 | 77.6 | 64.5 |
| Qwen2.5-VL-8B (Bai et al., 2025) | 57.1 | 60.8 | 26.9 | 58.0 | 66.7 | 53.9 |
| Qwen-Spatial-8B | 79.9(+22.8) | **78.7(+17.9)** | 34.4(+7.5) | 68.3(+10.3) | 80.0(+13.3) | 68.3(+14.4) |
| InternVL2.5-1B (Chen et al., 2024c) | 42.9 | 43.3 | 23.8 | 21.3 | 59.9 | 38.2 |
| InternVL-Spatial-1B | 65.4(+22.5) | 58.5(+15.2) | 26.3(+2.5) | 59.4(+28.1) | 74.4(+14.5) | 56.8(+18.6) |
| InternVL2.5-8B (Chen et al., 2024c) | 62.8 | 57.7 | 28.5 | 67.8 | 77.9 | 58.9 |
| InternVL-Spatial-8B | **87.8(+25.0)** | 78.6(+20.9) | 33.6(+5.1) | **71.3(+3.5)** | **83.9(+6.0)** | **71.0(+12.1)** |

To evaluate model performance in spatial reasoning, we conducted experiments on InternSpatial-Bench. The accuracy computation follows the methodology of Spatial-Bench (Cai et al., 2025) and Spatial-RPGT (Cheng et al., 2024), with a modification for the Object Counting task: since some VLMs struggle to follow instructions precisely, we extract the last number mentioned in the response as the predicted count and compute the relative error accordingly.

As shown in Table 2, our model, InternVL-Spatial-8B, outperforms the baseline InternVL2.5-8B (Chen et al., 2024c) by 12% in average accuracy. Notably, it achieves a 25% improvement in the Position Comparison task and a 20.9% gain in the Size Comparison task. Furthermore, InternVL-Spatial-8B surpasses advanced proprietary models such as GPT-4o (OpenAI, 2025) and Claude 3.5 Sonnet (Anthropic, 2024) across all tasks, demonstrating the effectiveness of InternSpatial in enhancing the spatial reasoning capabilities of VLMs.

To demonstrate the generality and broad impact of our InternSpatial, we applied the same training paradigm to two additional models with varying sizes: InternVL2.5-1B and Qwen2.5-VL-8B. The results consistently confirm the effectiveness of our training data. Specifically, InternVL-Spatial-1B improved its InternVL2.5-1B baseline by $18.6\%$ (from $38.2\%$ to $56.8\%$ average accuracy), exhibiting a significant lift for a smaller model. Similarly, the Qwen-Spatial-8B elevated the Qwen2.5-VL-8B baseline by $14.4\%$ (from $53.9\%$ to $68.3\%$ average accuracy). These substantial and consistent gains across different model families and sizes confirm that our proposed InternSpatial is highly effective and transferable for universally enhancing the spatial-aware capability of VLMs.

## 4.3 EVALUATION ON VSI-BENCH

To evaluate the additional multi-view spatial reasoning capabilities of InternVL-Spatial-8B trained on InternSpatial, we conducted experiments on VSI-Bench (Yang et al., 2024). As shown in Table 3, InternVL-Spatial-8B achieves notable improvements over the baseline InternVL2.5-8B (Chen et al., 2024c) across all tasks in the benchmark. In particular, it surpasses the baseline by more than 10% in Object Counting, Object Size Estimation, and Appearance Order tasks.

When compared against both open-source and proprietary models, InternVL-Spatial-8B delivers top-tier performance: it ranks first in Object Counting, Absolute Distance Estimation, Object Size

Table 3: Results on VSI-Bench. **Bold** indicates the best performance among all models, while underline denotes the second-best performance.

| Model | Obj.Count | Abs.Dist. | Obj.size | Room Size | Rel.Dist. | Route Plan | Appr.Order | Average |
|---|---|---|---|---|---|---|---|---|
| GPT-4o (OpenAI, 2025) | 46.2 | 5.3 | 43.8 | 38.2 | 37.0 | 31.5 | 28.5 | 32.9 |
| Gemini-1.5 Flash (Reid et al., 2024) | 49.8 | 30.8 | 53.5 | **54.4** | 37.7 | 31.5 | 37.8 | 42.3 |
| Gemini-1.5 Pro (Reid et al., 2024) | 56.2 | 30.9 | **64.1** | 43.6 | **51.3** | 36.0 | 34.6 | 45.3 |
| VILA-1.5-40B (Lin et al., 2024) | 22.4 | 24.8 | 48.7 | 22.7 | 40.5 | 31.5 | 32.9 | 32.0 |
| LLaVA-NeXT-Video-72B (Zhang et al., 2024b) | 48.9 | 22.8 | 57.4 | 35.3 | 42.4 | 35.0 | 48.6 | 41.5 |
| LLaVA-OneVision-72B (Li et al., 2024a) | 43.5 | 23.9 | 57.6 | 37.5 | 42.5 | 32.5 | 44.6 | 40.2 |
| Qwen2.5-VL-8B (Bai et al., 2025) | 41.5 | 21.2 | 50.7 | 36.6 | 37.9 | 30.4 | 34.0 | 36.0 |
| Qwen-Spatial-8B | 60.8(+19.3) | 35.0(+13.8) | 53.4(+2.7) | 45.0(+8.4) | 40.0(+2.1) | **36.6(+6.2)** | 34.5(+0.5) | 43.6(+7.6) |
| InternVL2.5-1B (Chen et al., 2024c) | 51.8 | 3.9 | 24.8 | 13.7 | 25.6 | 32.5 | 7.6 | 22.8 |
| InternVL-Spatial-1B | 66.4(+14.6) | 25.4(+21.5) | 42.0(+17.2) | 48.5(+24.8) | 34.1(+8.5) | 34.0(+1.5) | 11.0(+3.4) | 37.3(+14.5) |
| InternVL2.5-8B (Chen et al., 2024c) | 51.7 | 32.9 | 45.1 | 42.3 | 40.8 | 27.8 | 50.5 | 41.6 |
| InternVL-Spatial-8B | **68.7(+17.0)** | **40.9(+8.0)** | 63.1(+18.0) | 54.3(+12.0) | 47.7(+6.9) | 29.9(+2.1) | **60.5(+10.0)** | **52.3(+10.7)** |

Estimation and Appearance Order, and second in the remaining tasks. Overall, it achieves the highest average score among all evaluated models, including GPT-4o (OpenAI, 2025) and Gemini-1.5 Pro (Reid et al., 2024). These results demonstrate that InternSpatial substantially enhances the spatial reasoning capabilities of vision-language models in multi-image scenarios.

On VSI-BENCH, our InternSpatial yields consistent gains across architectures and scales. The InternVL-Spatial-1B gained a remarkable $14.5\%$ in average accuracy ($22.8\% \rightarrow 37.3\%$), with dramatic improvements in Absolute Distance ($+21.5\%$) and Room Size ($+24.8\%$). Similarly, the Qwen-Spatial-8B model elevated its baseline by a strong $7.6\%$ ($36.0\% \rightarrow 43.6\%$), including a $19.3\%$ gain in Object Counting. These results on a challenging spatial benchmark further show that InternSpatial can improve multi-view spatial reasoning in VLMs in a model-agnostic manner, without any architecture-specific modifications.

## 4.4 EFFECT OF THE VARIOUS QUESTION FORMATS

We conduct an ablation study on InternSpatial-Bench to evaluate the impact of different instruction formats in both the training step and evaluation step. Since the Rotation Estimation task does not include instruction format expansion, we exclude it from this analysis. Additionally, we train a variant of InternVL2.5-8B using InternSpatial-Bench without instruction format expansion, referred to as **InternVL-Spatial-Raw-8B**.

As shown in Figure 5, the baseline model, InternVL2.5-8B (Chen et al., 2024c), performs best on original images and natural language instructions, which are prevalent in general-purpose training datasets. However, it performs significantly worse on formats involving elements such as ¡box¿, which are rare in typical datasets. In contrast, InternVL-Spatial-8B, trained on InternSpatial with diverse instruction format expansions, substantially narrows this performance gap across different instruction styles. Furthermore, comparing InternVL2.5-8B with InternVL-Spatial-Raw-8B reveals that even without instruction format expansion, InternVL-Spatial-Raw-8B consistently outperforms the baseline across all instruction styles. This indicates that the model gains a degree of generalization and cross-format transfer ability, even without being explicitly trained on diverse instruction forms. Finally, InternVL-Spatial-8B achieves the best performance across all instruction formats, including natural language and original image styles. This demonstrates that instruction format expansion not only improves the model's robustness to diverse input styles but also enhances its overall spatial reasoning capability.

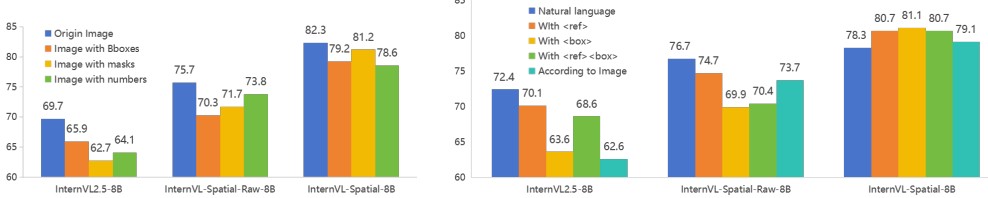

Figure 5: The results of the different image (**Left**) and text (**Right**) formats in the ablation study.

## 4.5 General VQA

For fairness, we re-evaluated InternVL2.5-8B (Chen et al., 2024c) under our experimental setup instead of directly using the results reported in its technical report. As shown in Table 4, InternVL-Spatial-8B achieves comparable performance to the baseline InternVL2.5-8B on general reasoning benchmarks. Specifically, InternVL-Spatial-8B shows a performance gain of +1.8% on Math-Vista (Wang et al., 2024b), -0.1% on OCRBench (Liu et al., 2024), +0.9% on TextVQA (Singh et al., 2019), -1.6% on ChartQA (Masry et al., 2022), and +0.2% on MMStar (Chen et al., 2024b). These results indicate that training with InternSpatial does not compromise the model's general reasoning capabilities, including mathematical reasoning, optical character recognition, VQA, and chart understanding.

Table 4: General benchmark results for InternVL2.5-8B vs. InternVL-Spatial-8B.

| Model | MathVision (Wang et al., 2024b) | OCRBench (Liu et al., 2024) | TextVQA (Singh et al., 2019) | ChartQA (Masry et al., 2022) | MMStar (Chen et al., 2024b) |
|---|---|---|---|---|---|
| InternVL2.5-8B | 19.0 | 82.3 | 79.0 | 83.0 | 62.9 |
| InternVL-Spatial-8B | 20.8(+1.8) | 82.2(-0.1) | 79.9(+0.9) | 81.4(-1.6) | 63.1(+0.2) |

## 5 Conclusions

We introduce InternSpatial, the largest open-source spatial reasoning dataset, and the benchmark InternSpatial-Bench, which together advance spatial understanding in VLMs through diverse scene coverage, rich instruction formats, and multi-view supervision. InternSpatial provides 12M high-quality QA pairs covering both single-view and multi-view settings, with broad scene diversity and 19 instruction formats that reflect the varied ways users express spatial queries. InternSpatial-Bench complements this with a diagnostic single-view benchmark and an extended multi-view evaluation via rotation angle prediction, a task not addressed in prior work. Extensive experiments show that training on InternSpatial yields substantial improvements on spatial reasoning benchmarks while maintaining strong performance on general multimodal tasks. Despite its scale and diversity, our template-based generation pipeline may underrepresent the full richness of natural language in real-world scenarios. Future work will explore more expressive QA generation and open-ended spatial reasoning in interactive environments. We anticipate that our dataset will support downstream applications such as robotics, embodied AI, and AR/VR, where spatial understanding is essential.

## Acknowledgements

This work was supported by the Shanghai Artificial Intelligence Laboratory, the Shanghai Committee of Science and Technology (No. 22YF1461500), and the National Natural Science Foundation of China (No. 62206046).

## Reproducibility Statement

All results reported in this paper are fully reproducible using the provided resources. The training configurations are detailed in Section 4 and Appendix E, while the dataset pipeline is described in Section 3, Appendix A, and Appendix B.

## Ethics Statement

Our work does not involve sensitive personal data. All dataset components were collected from open-source and publicly available sources, with careful filtering to exclude content that may be discriminatory or infringe copyright. We do not foresee any negative societal impacts arising from our methods or datasets.

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

APPENDIX

## A  EXPLANATION AND STATISTICS OF TASKS

InternSpatial and InternSpatial-Bench covers a total of 10 spatial reasoning tasks. The explanations of each task are shown in Table 5. We also count the number of QAs for each task in InternSpatial and InternSpatial-Bench, which are shown in Table 6 and Table 7 respectively.

Table 5: Explanation of tasks

| Task | Description |
|------|-------------|
| Position Comparison | Compare the position of two objects in an image, involving three pairs of positional relationship: left/right, above/below, near/far. |
| Size Comparison | Compare the size of two objects in an image, involving three pairs of size relationship: wider/thinner, taller/shorter, larger/smaller. |
| Existence Estimation | Determine whether there are objects in the image whose positional/size relationships with the specified object meet the constraint conditions. |
| Object Counting | Estimate how many objects that meet the constraint conditions there are in a single image or multiple images. |
| Rotation Estimation | Estimate the rotation angle of an object between two images. |
| Absolute Distance | Estimate the closest distance between two objects given a serial of images. |
| Room Size | Estimate the volume of the room(s) given a serial of images. |
| Object Size | Estimate the longest dimension of an object given a serial of images. |
| Route Plan | Given a serial of images, choose what action should be performed between a sequence of actions in order to route to from a start point to a target. |
| Appearance Order | Given a serial of images, determine the first-time appearance order of several objects. |

Table 6: Statistics of tasks in InternSpatial

| Task | Related Views | # of QAs |
|------|---------------|----------|
| Position Comparison | Single | 6,214,628 |
| Size Comparison | Single | 3,227,124 |
| Existence Estimation | Single | 50,845 |
| Object Counting | Single/Multiple | 53,866 |
| Rotation Estimation | Multiple | 2,464,500 |
| Absolute Distance | Multiple | 14,596 |
| Room Size | Multiple | 1,181 |
| Object Size | Multiple | 3,709 |
| Route Plan | Multiple | 4,966 |
| Appearance Order | Multiple | 8,562 |

## B  TEMPLATES FOR GENERATING QAS IN INTERNSPATIAL

The QAs in InternSpatial are generated by template-based generation method. Here we provide the full list of templates. The "[...]" in templates are placeholders which will be replaced by object references in different formats, values, choices, and so on. Several candidates are provided to be randomly selected in generation process to enrich the structure of sentences.

Table 7: Statistics of tasks in InternSpatial-Bench

| Task | Position Comparison | Size Comparison | Rotation Estimation | Object Counting | Existence Estimation |
|------|---------------------|-----------------|---------------------|-----------------|----------------------|
| # of QAs | 1845 | 1855 | 409 | 899 | 1000 |

Listing 1: Templates for task *Position Comparison*

```
above_predict_templates = {
  "question_templates": [
    "[A] is placed higher than [B], isn't it?",
    "Can we say that [A] is positioned above [B]?",
    "Is it correct to assume that [A] is located at a higher level than [
    B]?",
    "Is [A] placed higher than [B]?"
  ],
  "positive_answer_templates": [
    "Absolutely, [A] is clearly positioned above [B].",
    "Without a doubt, [A] is situated at a higher elevation than [B].",
    "Indeed, [A] is placed higher than [B].",
    "Certainly, [A] is located above [B]."
  ],
  "negative_answer_templates": [
    "Not at all, [A] is actually below [B].",
    "Definitely not, [A] is positioned lower than [B].",
    "Sorry, but [A] is not higher than [B].",
    "Unfortunately, [A] is not placed above [B]."
  ]
}
below_predict_templates = {
  "question_templates": [
    "[A] is placed lower than [B], right?",
    "Can we say that [A] is positioned below [B]?",
    "Is it correct to assume that [A] is situated lower than [B]?",
    "Is [A] placed lower than [B]?"
  ],
  "positive_answer_templates": [
    "Absolutely, [A] is clearly positioned below [B].",
    "Without a doubt, [A] is located lower than [B].",
    "Indeed, [A] is situated beneath [B].",
    "Certainly, [A] is found at a lower level than [B]."
  ],
  "negative_answer_templates": [
    "Not at all, [A] is actually higher than [B].",
    "Definitely not, [A] is positioned above [B].",
    "In fact, [A] is situated higher than [B].",
    "Quite the opposite, [A] is at a higher level than [B]."
  ]
}
left_predict_templates = {
  "question_templates": [
    "[A] is more to the left of [B], isn't it?",
    "Can we say that [A] is positioned more to the left than [B]?",
    "Is it correct to assume that [A] is situated to the left of [B]?",
    "Is [A] more to the left of [B]?"
  ],
  "positive_answer_templates": [
    "Absolutely, [A] is clearly positioned to the left of [B].",
    "Without a doubt, [A] is more to the left compared to [B].",
    "Indeed, [A] is located to the left of [B].",
    "Certainly, [A] is on the left side when compared to [B]."
  ],
  "negative_answer_templates": [
```

```
      "Not at all, [A] is not more to the left of [B].",
      "Actually, [A] is not positioned to the left of [B].",
      "Contrary to that, [A] is not situated to the left of [B].",
      "In fact, [A] is not on the left side when compared to [B]."
  ]
}
right_predict_templates = {
  "question_templates": [
    "[A] is more to the right of [B], isn't it?",
    "Can we say that [A] is positioned further to the right than [B]?",
    "Is it correct to assume that [A] is located to the right side of [B
    ]?",
    "Is [A] more to the right of [B]?"
  ],
  "positive_answer_templates": [
    "Absolutely, [A] is clearly positioned to the right of [B].",
    "Indeed, [A] is noticeably more to the right compared to [B].",
    "Without a doubt, [A] is situated further to the right than [B].",
    "Certainly, [A] is distinctly to the right of [B]."
  ],
  "negative_answer_templates": [
    "Not at all, [A] is actually to the left of [B].",
    "Definitely not, [A] is not positioned to the right of [B].",
    "In fact, [A] is on the left side of [B].",
    "Contrary to that, [A] is not further to the right than [B]."
  ]
}
near_predict_templates = {
  "question_templates": [
    "Is [A] positioned in front of [B]?",
    "Does [A] precede [B] in this arrangement?",
    "Is [A] in front of [B]?",
    "Is [A] closer to the observer than [B]?"
  ],
  "positive_answer_templates": [
    "Without a doubt, [A] stands nearer to the viewer than [B].",
    "Definitely, [A] is more proximate to the observer than [B].",
    "Indeed, [A] is in front of [B].",
    "Absolutely, [A] is before [B].",
  ],
  "negative_answer_templates": [
    "Not at all, [A] is not closer to the observer than [B].",
    "No, [A] is not in front of [B].",
    "Unfortunately, [A] is not ahead of [B].",
    "Definitely not, [A] is not closer to the observer than [B]."
  ]
}
far_predict_templates = {
  "question_templates": [
    "Is [A] situated behind [B]?",
    "Does [A] lie behind [B]?",
    "Is [A] to the rear of [B]?",
    "Is [A] farther from the observer than [B]?"
  ],
  "positive_answer_templates": [
    "Indeed, [A] is behind [B].",
    "Yes, [A] is behind [B].",
    "Without a doubt, [A] maintains a greater distance from the observer
    than [B].",
    "Certainly, [A] is positioned further away from the observer than [B
    ]."
  ],
  "negative_answer_templates": [
    "No, [A] is not behind [B].",
    "Incorrect, [A] is not behind [B].",
```

```
    "That's wrong. [A] is not positioned behind [B].",
    "Unfortunately, [A] is not to the rear of [B]."
  ]
}
above_choice_templates = {
  "question_templates": [
    "Which one is positioned at a higher elevation, [A] or [B]?",
    "In terms of altitude, which comes first, [A] or [B]?",
    "Who stands taller, [A] or [B]?",
    "Which is placed higher, [A] or [B]?"
  ],
  "answer_templates": [
    "[O] is the one that is placed higher.",
    "The higher position belongs to [O].",
    "[O] occupies the superior location.",
    "It is [O] that is situated at a greater height."
  ]
}
below_choice_templates = {
  "question_templates": [
    "Which is positioned closer to the ground, [A] or [B]?",
    "Which one is situated at a lower elevation, [A] or [B]?",
    "Which of these is nearer to the base level, [A] or [B]?",
    "Which is placed lower, [A] or [B]?"
  ],
  "answer_templates": [
    "[O] is placed lower.",
    "The lower position belongs to [O].",
    "[O] occupies the lower spot.",
    "Lower down, you'll find [O]."
  ]
}
left_choice_templates = {
  "question_templates": [
    "Which is positioned further to the left, [A] or [B]?",
    "In terms of leftward placement, which comes first, [A] or [B]?",
    "When considering the left side, which one is closer, [A] or [B]?",
    "Which is more to the left, [A] or [B]?"
  ],
  "answer_templates": [
    "[O] is located more to the left.",
    "The position of [O] is further to the left.",
    "In comparison, [O] stands out as being more on the left.",
    "It is evident that [O] is situated more towards the left."
  ]
}
right_choice_templates = {
  "question_templates": [
    "Which is positioned further to the right, [A] or [B]?",
    "In terms of horizontal alignment, which one is more to the right, [A
    ] or [B]?",
    "When comparing their positions, which one is situated more to the
    right, [A] or [B]?",
    "Which is more to the right, [A] or [B]?"
  ],
  "answer_templates": [
    "[O] is clearly more to the right.",
    "The position of [O] is further to the right.",
    "Comparing the two, [O] is definitively more to the right.",
    "It is evident that [O] is positioned more to the right."
  ]
}
near_choice_templates = {
  "question_templates": [
    "Which one is positioned further forward, [A] or [B]?",
```

```
    "Between [A] and [B], which object is closer to the observer?",
    "Can you identify which of the two, [A] or [B], is in the foremost
    position?",
    "Of the two, [A] and [B], which is closer to the front?"
  ],
  "answer_templates": [
    "[O] is in front.",
    "The frontmost object is [O].",
    "[O] is situated at the foremost position.",
    "Among the options, [O] is the one that is most ahead."
  ]
}
far_choice_templates = {
  "question_templates": [
    "Which one is further back, [A] or [B]?",
    "Can you tell me which is positioned more towards the back, [A] or [B
    ]?",
    "Between [A] and [B], which is more distant in the rear aspect?",
    "Comparing [A] and [B], which is more behind?"
  ],
  "answer_templates": [
    "[O] is definitely more behind.",
    "I can confirm that [O] is situated further back.",
    "[O] is clearly more behind than the other.",
    "There is no question that [O] is more behind."
  ]
}
above_below_choice_templates = {
  "question_templates": [
    "Is [A] positioned higher or lower than [B]?",
    "Does [A] lie above or beneath [B]?",
    "Is [A] situated over or under [B]?",
    "Is [A] above or below [B]?"
  ],
  "above_answer_templates": [
    "[A] is above [B].",
    "[A] is positioned higher than [B].",
    "[A] lies over [B].",
    "[A] is situated above [B]."
  ],
  "below_answer_templates": [
    "[A] is below [B].",
    "[A] is positioned lower than [B].",
    "[A] lies under [B].",
    "[A] is situated below [B]."
  ]
}
left_right_choice_templates = {
  "question_templates": [
    "Is [A] relatively farther to the left or right than [B]?",
    "Does [A] lie on the left or right side of [B]?",
    "Is [A] to the left or right of [B]?"
  ],
  "left_answer_templates": [
    "[A] is to the left of [B].",
    "[A] occupies the left side relative to [B].",
    "[A] lies on the left side of [B]."
  ],
  "right_answer_templates": [
    "[A] is to the right of [B].",
    "[A] occupies the right side relative to [B].",
    "[A] lies on the right side of [B]."
  ]
}
near_far_choice_templates = {
```

```
  "question_templates": [
    "Is [A] relatively nearer or farther from the observer than [B]?",
    "Can you determine if [A] is closer or farther from the observer
     compared to [B]?",
    "Is [A] in front of or behind [B]?"
  ],
  "near_answer_templates": [
    "[A] is closer to the observer than [B].",
    "[A] is more proximate to the observer than [B].",
    "[A] comes before [B].",
    "[A] is in front of [B]."
  ],
  "far_answer_templates": [
    "[A] is farther from the observer than [B].",
    "[A] is less proximate to the observer than [B].",
    "[A] is behind [B]."
  ]
}
```

Listing 2: Templates for task *Size Comparison*

```
wide_predict_templates = {
  "question_templates": [
    "Is [A] broader than [B]?",
    "Does [A] have a larger width compared to [B]?",
    "Can we say that [A] spans more horizontally than [B]?",
    "Is [A] wider than [B]?"
  ],
  "positive_answer_templates": [
    "Yes, [A] is noticeably broader than [B].",
    "Indeed, [A] has a significantly larger width than [B].",
    "Absolutely, [A] spans more horizontally than [B].",
    "Certainly, [A] is wider than [B]."
  ],
  "negative_answer_templates": [
    "No, [A] is not broader than [B].",
    "In fact, [A] does not have a larger width compared to [B].",
    "Sorry, but [A] does not span more horizontally than [B].",
    "Unfortunately, [A] is not wider than [B]."
  ]
}
narrow_predict_templates = {
  "question_templates": [
    "Is [A] thinner than [B]?",
    "Does [A] have a smaller width compared to [B]?",
    "Is the width of [A] less than that of [B]?",
    "Is [A] narrower than [B]?"
  ],
  "positive_answer_templates": [
    "Yes, [A] is noticeably narrower than [B].",
    "Indeed, [A] has a significantly smaller width than [B].",
    "Absolutely, the width of [A] is less than that of [B].",
    "Certainly, [A] is thinner than [B]."
  ],
  "negative_answer_templates": [
    "No, [A] is not narrower than [B]; in fact, it's wider.",
    "Definitely not; [A] has a larger width than [B].",
    "Not at all; the width of [A] exceeds that of [B].",
    "No way; [A] is thicker than [B]."
  ]
}
tall_predict_templates = {
  "question_templates": [
    "[A] is taller than [B], isn't it?",
    "Can we say that [A] surpasses [B] in height?",
```

```
      "Is it correct to assume that [A] is taller than [B]?",
      "Is [A] taller than [B]?"
    ],
    "positive_answer_templates": [
      "Absolutely, [A] towers over [B].",
      "Without a doubt, [A] is significantly taller than [B].",
      "Indeed, [A] outshines [B] in terms of height.",
      "Unquestionably, [A] is taller than [B]."
    ],
    "negative_answer_templates": [
      "Not at all, [B] is actually taller than [A].",
      "Sorry, but [A] does not exceed [B] in height.",
      "In fact, [B] surpasses [A] in height.",
      "Regrettably, [A] falls short when compared to [B]'s height."
    ]
}
vshort_predict_templates = {
    "question_templates": [
      "Is [A] shorter than [B] in vertical direction?",
      "Does [A] have less height than [B]?",
      "Is the vertical length of [A] smaller than that of [B]?",
      "Is [A] shorter than [B] in vertical direction?"
    ],
    "positive_answer_templates": [
      "Yes, [A] is indeed shorter than [B] in the vertical direction.",
      "Absolutely, [A] has less height compared to [B].",
      "Certainly, the vertical length of [A] is smaller than that of [B].",
      "Without a doubt, [A] is shorter than [B] vertically."
    ],
    "negative_answer_templates": [
      "No, [A] is not shorter than [B] in the vertical direction.",
      "Definitely not, [A] does not have less height than [B].",
      "Not at all, the vertical length of [A] is not smaller than that of [
    B].",
      "Certainly not, [A] is not shorter than [B] vertically."
    ]
}
large_predict_templates = {
    "question_templates": [
      "[A] is larger than [B], isn't it?",
      "Can we say that [A] has a bigger size compared to [B]?",
      "Is it correct to assume that [A] surpasses [B] in size?",
      "Is [A] larger than [B]?"
    ],
    "positive_answer_templates": [
      "Absolutely, [A] is noticeably larger than [B].",
      "Without a doubt, [A] outsizes [B] significantly.",
      "Indeed, [A] is clearly more expansive than [B].",
      "Definitely, [A] dwarfs [B] in terms of size."
    ],
    "negative_answer_templates": [
      "Not at all, [B] is actually larger than [A].",
      "Quite the opposite, [B] surpasses [A] in size.",
      "In fact, [B] is the larger one when compared to [A].",
      "Sorry, but [B] is bigger than [A]."
    ]
}
small_predict_templates = {
    "question_templates": [
      "[A] is smaller than [B], isn't it?",
      "Can we say that [A] is smaller than [B]?",
      "Is it true that [A] is smaller than [B]?",
      "Is [A] smaller than [B]?"
    ],
    "positive_answer_templates": [
```

```
    "Absolutely, [A] is noticeably smaller than [B].",
    "Yes, [A] is indeed smaller than [B].",
    "Without a doubt, [A] is smaller than [B].",
    "Definitely, [A] is smaller than [B]."
  ],
  "negative_answer_templates": [
    "Not at all, [A] is actually larger than [B].",
    "No, [A] is not smaller than [B].",
    "Quite the opposite, [A] is bigger than [B].",
    "False, [A] is not smaller than [B]."
  ]
}
wide_choice_templates = {
  "question_templates": [
    "Which has a greater width, [A] or [B]?",
    "In terms of width, which one is larger, [A] or [B]?",
    "When comparing widths, which one comes out on top, [A] or [B]?",
    "Which is wider, [A] or [B]?"
  ],
  "answer_templates": [
    "[O] is wider.",
    "The width of [O] is greater.",
    "Comparing the two, [O] has the larger width.",
    "In terms of width, [O] surpasses the other."
  ]
}
narrow_choice_templates = {
  "question_templates": [
    "Which has a smaller width, [A] or [B]?",
    "In terms of width, which one is less, [A] or [B]?",
    "When comparing widths, which one comes out smaller, [A] or [B]?",
    "Which is narrower, [A] or [B]?"
  ],
  "answer_templates": [
    "[O] is the narrower one.",
    "The narrower object is [O].",
    "[O] has the lesser width.",
    "Comparing the two, [O] is clearly narrower."
  ]
}
tall_choice_templates = {
  "question_templates": [
    "Which has a greater height, [A] or [B]?",
    "In terms of height, which one is superior, [A] or [B]?",
    "When comparing heights, which comes out on top, [A] or [B]?",
    "Which is taller, [A] or [B]?"
  ],
  "answer_templates": [
    "[O] is the taller one.",
    "The height of [O] surpasses the other.",
    "[O] stands out as the taller between the two.",
    "Comparatively speaking, [O] is taller."
  ]
}
vshort_choice_templates = {
  "question_templates": [
    "Which has a shorter vertical length, [A] or [B]?",
    "In terms of vertical measurement, which one is shorter, [A] or [B]?"
    ,
    "When comparing the vertical dimensions, which is shorter, [A] or [B
    ]?",
    "Which is shorter in vertical direction, [A] or [B]?"
  ],
  "answer_templates": [
    "[O] is shorter in the vertical direction.",
```

```
      "The vertical length of [O] is less than the other.",
      "Comparing vertically, [O] comes out shorter.",
      "In terms of height, [O] is the shorter one."
   ]
}
large_choice_templates = {
   "question_templates": [
      "Which has a greater size, [A] or [B]?",
      "In terms of size, which one is bigger, [A] or [B]?",
      "When comparing sizes, which one comes out on top, [A] or [B]?",
      "Which is larger, [A] or [B]?"
   ],
   "answer_templates": [
      "[O] is the larger one.",
      "The bigger size belongs to [O].",
      "[O] surpasses the other in size.",
      "Comparatively speaking, [O] is the larger."
   ]
}
small_choice_templates = {
   "question_templates": [
      "Which has a smaller size, [A] or [B]?",
      "In terms of size, which one is smaller, [A] or [B]?",
      "When comparing sizes, which one comes out smaller, [A] or [B]?",
      "Which is smaller, [A] or [B]?"
   ],
   "answer_templates": [
      "[O] is the smaller one.",
      "The smaller of the two is [O].",
      "[O] has the smaller size.",
      "Comparing the two, [O] is the smaller."
   ]
}
wide_narrow_choice_templates = {
   "question_templates": [
      "Is [A] relatively wider or narrower than [B]?",
      "How does the width of [A] compare to [B]?",
      "Can you tell me if [A] has a greater or lesser width than [B]?",
      "Is [A] wider or narrower than [B]?"
   ],
   "wide_answer_templates": [
      "[A] is wider than [B].",
      "The width of [A] exceeds that of [B].",
      "[A] has a larger width compared to [B].",
      "In terms of width, [A] surpasses [B]."
   ],
   "narrow_answer_templates": [
      "[A] is narrower than [B].",
      "The width of [B] is greater than that of [A].",
      "[A] has a smaller width compared to [B].",
      "In terms of width, [B] surpasses [A]."
   ]
}
tall_short_choice_templates = {
   "question_templates": [
      "Is [A] relatively taller or shorter than [B]?",
      "How does the height of [A] compare to [B]?",
      "Can you determine if [A] is taller or shorter than [B]?",
      "Is [A] taller or shorter than [B]?"
   ],
   "tall_answer_templates": [
      "[A] is taller than [B].",
      "The height of [A] exceeds that of [B].",
      "[A] surpasses [B] in height.",
      "Compared to [B], [A] is definitely taller."
```

```
    ],
    "short_answer_templates": [
      "[A] is shorter than [B].",
      "In terms of height, [A] falls below [B].",
      "[B] is taller than [A].",
      "[A]'s height is less than that of [B]."
    ]
}
large_small_choice_templates = {
    "question_templates": [
      "Is [A] relatively larger or smaller than [B]?",
      "How does the size of [A] compare to [B]?",
      "Can you determine if [A] is bigger or smaller than [B]?",
      "Is [A] larger or smaller than [B]?"
    ],
    "large_answer_templates": [
      "[A] is larger than [B].",
      "The size of [A] exceeds that of [B].",
      "[A] surpasses [B] in size.",
      "Compared to [B], [A] is bigger."
    ],
    "small_answer_templates": [
      "[A] is smaller than [B].",
      "The size of [A] is less than that of [B].",
      "[B] is larger than [A].",
      "In comparison to [B], [A] is smaller."
    ]
}
```

Listing 3: Templates for task *Existence Estimation*

```
existence_left_templates = {
    "question_templates": [
      "Does [B] exist to the left of [A]?",
      "Is there [B] to the left of [A]?",
      "Is there [B] more to the left than [A]?"
    ],
    "positive_answer_templates": [
      "Yes."
    ],
    "negative_answer_templates": [
      "No."
    ]
}
existence_right_templates = {
    "question_templates": [
      "Is there [B] positioned more to the right than [A]?",
      "Does [B] exist to the right of [A]?",
      "Is there [B] locating to the rightside of [A]?"
    ],
    "positive_answer_templates": [
      "Yes."
    ],
    "negative_answer_templates": [
      "No."
    ]
}
existence_above_templates = {
    "question_templates": [
      "Does [B] exist at higher elevation than [A]?",
      "Can you find [B] above [A]?",
      "Is there [B] that is located above [A]?"
    ],
    "positive_answer_templates": [
      "Yes."
```

```
    ],
    "negative_answer_templates": [
      "No."
    ]
}
existence_below_templates = {
    "question_templates": [
      "Is there [B] that is situated below [A]?",
      "Does [B] exist below [A]?",
      "Is there [B] positioned lower than [A]?"

    ],
    "positive_answer_templates": [
      "Yes."
    ],
    "negative_answer_templates": [
      "No."
    ]
}
existence_near_templates = {
    "question_templates": [
      "Does [B] exist near [A]?",
      "Is there [B] that is in front of [A]?",
      "Can you find [B] that is closer than observer than [A]?"
    ],
    "positive_answer_templates": [
      "Yes."
    ],
    "negative_answer_templates": [
      "No."
    ]
}
existence_far_templates = {
    "question_templates": [
      "Does [B] exist far from [A]?",
      "Is there [B] that is behind [A]?",
      "Does [B] exist behind [A]?"
    ],
    "positive_answer_templates": [
      "Yes."
    ],
    "negative_answer_templates": [
      "No."
    ]
}
existence_wide_templates = {
    "question_templates": [
      "Can you find [B] that is wider than [A]?",
      "Is there [B] that is wider than [A]?",
      "Is there [B] that has a larger extent in horizontal than [A]?"

    ],
    "positive_answer_templates": [
      "Yes."
    ],
    "negative_answer_templates": [
      "No."
    ]
}
existence_narrow_templates = {
    "question_templates": [
      "Is there [B] that is narrower than [A]?",
      "Can you find [B] that is narrower than [A]?",
      "Does [B] with smaller width than [A] exist?"
    ],
```

```
  "positive_answer_templates": [
    "Yes."
  ],
  "negative_answer_templates": [
    "No."
  ]
}
existence_tall_templates = {
  "question_templates": [
    "Is there [B] that is taller than [A]?",
    "Can you find [B] that has a larger height than [A]?"
    "Is there [B] that is larger in vertical than [A]?"
  ],
  "positive_answer_templates": [
    "Yes."
  ],
  "negative_answer_templates": [
    "No."
  ]
}
existence_vshort_templates = {
  "question_templates": [
    "Is there [B] that is shorter than [A] in vertical?",
    "Does [B] shorter than [A] exists?",
    "Is there [B] that has a smaller height than [A]?"
  ],
  "positive_answer_templates": [
    "Yes."
  ],
  "negative_answer_templates": [
    "No."
  ]
}
existence_large_templates = {
  "question_templates": [
    "Can you find [B] that is larger than [A]?",
    "Is there [B] that is larger than [A]?",
    "Does [B] exist that has a larger volume than [A]?"
  ],
  "positive_answer_templates": [
    "Yes."
  ],
  "negative_answer_templates": [
    "No."
  ]
}
existence_small_templates = {
  "question_templates": [
    "Is there [B] that is smaller in size than [A]?",
    "Does [B] with smaller size than [A] exist?",
    "Does [B] exist that is smaller than [A]?"
  ],
  "positive_answer_templates": [
    "Yes."
  ],
  "negative_answer_templates": [
    "No."
  ]
}
```

Listing 4: Templates for task *Object Counting*

```
count_above_templates = {
    "question_templates": [
        "How many [B] are located higher than [A]?",
```

```
            "How many [B] are positioned higher than [A]?",
            "How many [B] are above [A]?"
        ],
        "answer_templates": [
            "[V]."
        ]
}
count_below_templates = {
        "question_templates": [
            "How many [B] are lower than [A]?",
            "How many [B] are situated below [A]?",
            "How many [B] are positioned lower than [A]?"
        ],
        "answer_templates": [
            "[V]."
        ]
}
count_left_templates = {
        "question_templates": [
            "How many [B] are positioned to the left of [A]?",
            "How many [B] are more to the left than [A]?",
            "How many [B] are on the leftside of [A]?"
        ],
        "answer_templates": [
            "[V]."
        ]
}
count_right_templates = {
        "question_templates": [
            "How many [B] are found to the right of [A]?",
            "How many [B] lie to the rightside of [A]?",
            "How many [B] are more to the right than [A]?"
        ],
        "answer_templates": [
            "[V]."
        ]
}
count_near_templates = {
        "question_templates": [
            "How many [B] are closer to the observer than [A]?",
            "How many [B] are in front of [A]?",
            "How many [B] are located nearer to the observer than [A]?"
        ],
        "answer_templates": [
            "[V]."
        ]
}
count_far_templates = {
        "question_templates": [
            "How many [B] are positioned farther from the observer than [A]?"
        ,
            "How many [B] are located behind [A]?"
            "How many [B] are farther from the observer than [A]?"
        ],
        "answer_templates": [
            "[V]."
        ]
}
count_wide_templates = {
        "question_templates": [
            "How many [B] have a larger width compared to [A]?",
            "How many [B] are wider than [A]?",
            "How many [B] have a larger extent in horizontal than [A]?"
        ],
        "answer_templates": [
```

```
            "[V]."
    ]
}
count_narrow_templates = {
    "question_templates": [
        "How many [B] are narrower than [A]?",
        "How many [B] have a less width than that of [A]?",
        "How many [B] are thinner than [A]?"
    ],
    "answer_templates": [
        "[V]."
    ]
}
count_tall_templates = {
    "question_templates": [
        "How many [B] are taller than [A]?",
        "How many [B] surpass [A] in height?",
        "How many [B] have a larger extent in vertical than [A]?"
    ],
    "answer_templates": [
        "[V]."
    ]
}
count_vshort_templates = {
    "question_templates": [
        "How many [B] have less height than [A]?",
        "How many [B] are shorter than [A]?",
        "How many [B] have a smaller vertical length than that of [A]?"
    ],
    "answer_templates": [
        "[V]."
    ]
}
count_large_templates = {
    "question_templates": [
        "How many [B] are larger than [A]?",
        "How many [B] have a bigger size compared to [A]?",
        "How many [B] surpass [A] in size?"
    ],
    "answer_templates": [
        "[V]."
    ]
}
count_small_templates = {
    "question_templates": [
        "How many [B] have a smaller size compared to [A]?"
        "How many [B] are smaller than [A]?",
        "How many [B] are smaller in volume than [A]?"
    ],
    "answer_templates": [
        "[V]."
    ]
}
```

Listing 5: Templates for multi-view tasks

```
object_rotation_predict_templates = {
  "question_templates": [
    "Here are two images of the same object:\nImage 1:\n<image>\nImage
    2:\n<image>\nPlease estimate how [A] in image 2 is rotated relative
    to image 1?",
    "Here are two images of the same object:\nImage 1:\n<image>\nImage
    2:\n<image>\nIn what direction and by what angle has [A] in image 2
    been rotated from its position in image 1?"
  ],
```

```
  "clockwise_answer_templates": [
    "[A] rotates about [D] degrees clockwise.",
    "[A] turns clockwise by about [D] degrees.",
    "[A] undergoes approximately a [D] degree clockwise rotation."
  ],
  "counterclockwise_answer_templates": [
    "[A] rotates abount [D] degrees counterclockwise.",
    "[A] turns counterclockwise by about [D] degrees.",
    "[A] undergoes approximately a [D] degree counterclockwise rotation."
  ],
  "rotate_180_answer_templates": [
    "[A] rotates about [D] degrees.",
    "[A] turns by about [D] degrees.",
    "[A] undergoes approximately a [D] degree rotation."
  ]
}
route_plan_templates = {
  "question_templates": [
    "Image-1: <image>\nImage-2: <image>\nImage-3: <image>\nImage-4: <
    image>\nImage-5: <image>\nImage-6: <image>\nImage-7: <image>\nImage
    -8: <image>\nImage-9: <image>\nImage-10: <image>\nImage-11: <image>\
    nImage-12: <image>\nImage-13: <image>\nImage-14: <image>\nImage-15: <
    image>\nImage-16: <image>\nImage-17: <image>\nImage-18: <image>\
    nImage-19: <image>\nImage-20: <image>\nImage-21: <image>\nImage-22: <
    image>\nImage-23: <image>\nImage-24: <image>\nYou are a robot
    beginning at the column and facing the staircase. You want to
    navigate to the grand staircase. You will perform the following
    actions (Note: for each [please fill in], choose either 'turn back,'
    'turn left,' or 'turn right.'): 1. Go forward until the columns. 2. [
    please fill in]. 3. Go forward until the steps. 4. Stop on the
    landing.\nA. Turn Right\nB. Turn Left\nC. Turn Back\nAnswer with the
    option's letter from the given choices directly."
  ],
  "answer_templates": [
    "[O]"
  ]
}
abs_dist_templates = {
  "question_templates": [
    "Image-1: <image>\nImage-2: <image>\nImage-3: <image>\nImage-4: <
    image>\nImage-5: <image>\nImage-6: <image>\nImage-7: <image>\nImage
    -8: <image>\nImage-9: <image>\nImage-10: <image>\nImage-11: <image>\
    nMeasuring from the closest point of each object, what is the
    distance between [A] and [B] (in meters)?\nPlease answer the question
     using a single word or phrase."
  ],
  "answer_templates": [
    "[V]"
  ]
}
obj_count_templates = {
  "question_templates": [
    "Image-1: <image>\nImage-2: <image>\nImage-3: <image>\nImage-4: <
    image>\nImage-5: <image>\nImage-6: <image>\nImage-7: <image>\nImage
    -8: <image>\nImage-9: <image>\nImage-10: <image>\nImage-11: <image>\
    nThese are frames of a video.\nHow many [A](s) are in this room?\
    nPlease answer the question using a single word or phrase."
  ],
  "answer_templates": [
    "[V]"
  ]
}
room_size_templates = {
  "question_templates": [
```

```
    "Image-1: <image>\nImage-2: <image>\nImage-3: <image>\nImage-4: <
    image>\nImage-5: <image>\nImage-6: <image>\nImage-7: <image>\nImage
    -8: <image>\nImage-9: <image>\nImage-10: <image>\nImage-11: <image>\
    nThese are frames of a video.\nWhat is the size of this room (in
    square meters)? \nIf multiple rooms are shown, estimate the size of
    the combined space.\nPlease answer the question using a single word
    or phrase."
  ],
  "answer_templates": [
    "[V]"
  ]
}
rel_dist_templates = {
  "question_templates": [
    "Image-1: <image>\nImage-2: <image>\nImage-3: <image>\nImage-4: <
    image>\nImage-5: <image>\nImage-6: <image>\nImage-7: <image>\nImage
    -8: <image>\nImage-9: <image>\nImage-10: <image>\nThese are frames of
     a video.\nMeasuring from the closest point of each object, which of
    these objects ([B0],[B1],[B2],[B3]) is the closest to [A]?\nA. [B0]\
    nB. [B1]\nC. [B2]\nD. [B3]\nAnswer with the option's letter from the
    given choices directly."
  ],
  "answer_templates": [
    "[O]"
  ]
}
object_size_templates = {
  "question_templates": [
    "Image-1: <image>\nImage-2: <image>\nImage-3: <image>\nImage-4: <
    image>\nImage-5: <image>\nImage-6: <image>\nImage-7: <image>\nImage
    -8: <image>\nImage-9: <image>\nImage-10: <image>\nImage-11: <image>\
    nThese are frames of a video.\nWhat is the length of the longest
    dimension (length, width, or height) of [A], measured in centimeters
    ?\nPlease answer the question using a single word or phrase."
  ],
  "answer_templates": [
    "[V]"
  ]
}
appear_order_templates = {
  "question_templates": [
    "Image-1: <image>\nImage-2: <image>\nImage-3: <image>\nImage-4: <
    image>\nImage-5: <image>\nImage-6: <image>\nImage-7: <image>\nImage
    -8: <image>\nImage-9: <image>\nImage-10: <image>\nImage-11: <image>\
    nThese are frames of a video.\nWhat will be the first-time appearance
     order of the following categories in the video: [A0], [A1], [A2], [
    A3]?\nA. [B0]\nB. [B1]\nC. [B2]\nD. [B3]\nAnswer with the option's
    letter from the given choices directly."
  ],
  "answer_templates": [
    "[O]"
  ]
}
```

## C  VLM-ASSISTED ANNOTATION FOR INTERNSPATIAL

As described in Dataset section, we involved open-source VLM to do the object detection, captioning, and grounding in the pipeline of InternSpatial generation. We use QWen2.5-VL 72B(Bai et al., 2025) as the assistant. For each process, we design corresponding prompt to make the VLM understand what should do and what should output. Here we provide the prompts for these processes.

Listing 6: Prompts for detecting objects in images

```
messages = [{"role": "system", "content": f"""
```

```
  You are an object detector. Given an image, you should find all objects
     in image with grounding. The term "object" includes all living and
     non-living things. For each detected object, you should assign a
     label, which represents what the object is. You should also describe
     each detected object in detail with a phrase. The description can
     contain appearance, function, action, etc.

  Output format: The response should be in json format, which contains a
     list of dicts. Each dict is for an detected object and has three keys
     : "label" for the label, "caption" for the description and "box" for
     the grounding. The description should be lowercases and no period at
     the end. The grounding should be a list of four ints [x1, y1, x2, y2
     ], where (x1, y1) is the top-left coord and (x2, y2) is the bottom-
     right coord. Compact the responsed json in one line.
"""}]
messages.append({"role": "user", "content": '\n'.join(query)})
```

Listing 7: Prompts for captioning objects given bounding boxes

```
messages = [{"role": "system", "content": f"""
  You are an language assistant. You will be given an array dict. Each
     dict contains a field "box" for grounding box and an optional field "
     label" for label of a object in image. Your task is to generate brief
      descriptions with less than ten words for these objects. Output one
     description per line.

  Here is an example:

  Input:
  [{"box": [10, 20, 300, 400], "label": "bus"}, {"box": [42, 512, 64,
     890]}]

  Output:
  a blue bus seat with a suitcase partially resting on it
  a red car on right side
"""}]
messages.append({"role": "user", "content": '\n'.join(query)})
```

Listing 8: Prompts for grounding objects given captions

```
messages = [{"role": "system", "content": f"""
  You are a professional image annotator. I will give you an image and a
     phrase about one or more objects in the image. Please detect all
     objects matching the phrase. The response should be a JSON object,
     containing a field "boxes". "boxes" is a list of grounding boxes [x1,
      y1, x2, y2].

  Example Output:
  {
    "boxes": [[192, 29, 321, 49], [19, 65, 392, 569], [59, 102, 439,
     139]]
  }
"""}]
messages.append({"role": "user", "content": '\n'.join(query)})
```

## D MORE DETAILS ABOUT INTERNSPATIAL-BENCH GENERATION PIPELINE

The generation pipeline of InternSpatial-Bench does not rely on existing annotations. Starting from the images, we carried out four steps including image filtering, image captioning, question design, and object grounding to obtain the necessary 2D annotations for the questions of the benchmark and the generation of answers. In this steps, we design prompts respectively to enable the Visual Language Model (VLM) to automatically generate intermediate results. These prompts are presented in Listing 9, Listing 10, Listing 11, Listing 12, Listing 13, and Listing 8. Subsequently, we reused

the processes in the second and third stages of the training dataset pipeline to generate answers and expand the instruction formats. After generated the QA pairs, we invited experienced human annotators to conduct manual verification of all the pairs to ensure the quality of the benchmark.

Listing 9: Prompts for filtering images

```
messages = [{"role": "system", "content": f"""
  You are a helpful visual assistant. Please determine whether the
    following conditions are met:
  1. 5-or-more-objects: There are at least 5 objects in the image.
  2. cartoon: This is a cartoon image.
  3. group-of-images: This is a group of images.
  4. screenshot: This is a screenshot.
  5. realistic-image: This is a realistic image captured by camera.

  For each condition, answer true of false. Response in JSON dict with
    five fields: "5-or-more-objects", "cartoon", "group-of-images", "
    screenshot", "realistic-image
"""}]
messages.append({"role": "user", "content": '\n'.join(query)})
```

Listing 10: Prompts for captioning images

```
messages = [{"role": "system", "content": f"""
  You are a helpful visual assistant. Please describe the image as detail
     as possible. Then detect all top-level objects and return their
    detailed descriptions (top-level means it's not a part of another
    object).
"""}]
messages.append({"role": "user", "content": '\n'.join(query)})
```

Listing 11: Prompts for design questions of task *Position Comparison*

```
messages = [{"role": "system", "content": f"""
  I will give you an image and a description about the image. You should
    design 2 questions regarding position judgments around top-level
    objects in the image (top-level means it's not a part of another
    object). The question should involve two object (anchor, target) and
    a type of relationship:

  - *The anchor and target object* should be randomly chosen from the top
    -level objects. You possibly need to add the attributions about
    appearance, behavior, posture, position in the image, etc. to the
    description of the anchor and target objects so that they can be
    distinguished from others.
  - *The relationship* should be randomly selected from: more to the left
    , more to the right, closer (to the observer), farther (from the
    observer), higher, lower.
  - Only design questions about top-level objects. Ignore those not in
    top-level objects list. Ignore environment objects such as water, sky
    , grass, cloud, etc.

  After that, generate 2 more questions based on the designed questions
    by choose another relationship and keep other parts unchanged.

  Please respond in JSON format. All content should be in English. Here
    is an example of output:
  {
    "questions": [
      {
        "question": "Is the wooden chair positioned higher than the blue
    table?",
        "anchor": "blue table",
        "target": "wooden chair",
        "relationship": "higher",
        "task": "position"
```

```
      },
      {
        "question": "Does the red bicycle locate more to the left than
    the man in a floral shirt?",
        "anchor": "man in a floral shirt",
        "target": "red bicycle",
        "relationship": "more left",
        "task": "position"
      }
    ],
    "modified_questions": [
      {
        "question": "Is the wooden chair farther from the observer than
    the blue table?",
        "anchor": "blue table",
        "target": "wooden chair",
        "relationship": "farther",
        "task": "position"
      },
      {
        "question": "Is the red bicycle located at a lower elevation than
     the man in a floral shirt?",
        "anchor": "man in a floral shirt",
        "target": "red bicycle",
        "relationship": "lower",
        "task": "position"
      }
    ]
  }
"""}]
messages.append({"role": "user", "content": '\n'.join(query)})
```

Listing 12: Prompts for design questions of task *Size Comparison*

```
messages = [{"role": "system", "content": f"""
  I will give you an image and a description about the image. You should
    design 2 questions regarding size judgments around top-level objects
    in the image (top-level means it's not a part of another object). The
     question should involve two object (anchor, target) and a type of
    relationship:

  - *The anchor and target object* should be randomly chosen from the top
    -level objects. You possibly need to add the attributions about
    appearance, behavior, posture, position in the image, etc. to the
    description of the anchor and target objects so that they can be
    distinguished from others.
  - *The relationship* should be randomly selected from: larger, smaller,
     taller, shorter, wider, narrower.
  - Only design questions about top-level objects. Ignore those not in
    top-level objects list. Ignore environment objects such as water, sky
    , grass, cloud, etc.

  After that, generate 2 more questions based on the designed questions
    by choose another relationship and keep other parts unchanged.

  Please respond in JSON format. All content should be in English. Here
    is an example of output:
  {
    "questions": [
      {
        "question": "Is the green vase taller than the brown table?",
        "anchor": "brown table",
        "target": "green vase",
        "relationship": "taller",
        "task": "size"
```

```
      },
      {
        "question": "Is the plate with food on it narrower than the white
   box in the middle?",
        "anchor": "white box in the middle",
        "target": "plate with food on it",
        "relationship": "narrower",
        "task": "size"
      }
    ],
    "modified_questions": [
      {
        "question": "Is the green vase smaller than the brown table?",
        "anchor": "brown table",
        "target": "green vase",
        "relationship": "smaller",
        "task": "size"
      },
      {
        "question": "Is the plate with food on it larger than the white
   box in the middle?",
        "anchor": "white box in the middle",
        "target": "plate with food on it",
        "relationship": "larger",
        "task": "size"
      }
    ]
  }
"""}]
messages.append({"role": "user", "content": '\n'.join(query)})
```

Listing 13: Prompts for design questions of task *Existence Estimation* and *Object Counting*

```
messages = [{"role": "system", "content": f"""
  I will give you an image and a description about the image. You should
    design 2 questions regarding existence judgments and 2 questions
    regarding counting around top-level objects in the image (top-level
    means it's not a part of another object). The conditions in the
    question need to involve an anchor object and a type of relationship:

  - *The anchor object* should be randomly chosen from the top-level
    objects. If multiple objects in the image are similar to anchor
    object, you need to add the attributions about appearance, behavior,
    posture, position in the image, etc. to the description of anchor
    object so that it can be distinguished from others.
  - *The relationship* should randomly selected from: more to the left,
    more to the right, closer (to the observer), farther (to the observer
    ), higher, lower, larger, smaller, taller, shorter, wider, narrower.
  - Only design questions about top-level objects. Ignore those not in
    top-level objects list. Ignore environment objects such as water, sky
    , grass, cloud, etc.

  After that, generate 4 more questions based on the designed questions
    by choose another type of relationship and keep other parts unchanged
    .

  Please respond in JSON format. All content should be in English. Here
    is an example of output:
  {
    "questions": [
      {
        "question": "Are there chairs wider than the blue table?",
        "anchor": "blue table",
        "target": "chair",
        "relationship": "wider",
```

```
        "task": "existence"
      },
      {
        "question": "Is there a bicycle located more to the left than the
    man in a floral shirt?",
        "anchor": "man in a floral shirt",
        "target": "bicycle",
        "relationship": "more left",
        "task": "existence"
      },
      {
        "question": "How many green vases are positioned higher than the
    middle wooden table?",
        "anchor": "middle wooden table",
        "target": "green vase",
        "relationship": "higher",
        "task": "count"
      },
      {
        "question": "How many plates are larger than the white box in the
    middle?",
        "anchor": "white box in the middle",
        "target": "plate",
        "relationship": "larger",
        "task": "count"
      }
    ],
    "modified_questions": [
      {
        "question": "Are there chairs which have lower elevation than the
    blue table?",
        "anchor": "blue table",
        "target": "chair",
        "relationship": "lower",
        "task": "existence"
      },
      {
        "question": "Is there a bicycle closer to the observer than the
    man in a floral shirt?",
        "anchor": "man in a floral shirt",
        "target": "bicycle",
        "relationship": "closer",
        "task": "existence"
      },
      {
        "question": "How many green vases are wider than the middle
    wooden table?",
        "anchor": "middle wooden table",
        "target": "green vase",
        "relationship": "wider",
        "task": "count"
      },
      {
        "question": "How many plates are shorter than the white box in
    the middle?",
        "anchor": "white box in the middle",
        "target": "plate",
        "relationship": "shorter",
        "task": "count"
      }
    ]
  }
"""}]
messages.append({"role": "user", "content": '\n'.join(query)})
```

# E  TRAINING DETAILS

We train InternVL-Spatial-8B using LoRA(Hu et al., 2022) with approximately 291K general training samples from InternVL2.5 (Chen et al., 2024c) and 2M samples from InternSpatial, counted with repetition. The training is conducted on 16 A100 GPUs for approximately 14 hours. We report the models and training hyperparameters of InternVL-Spatial-8B in Table 8.

Table 8: **Training settings and hyperparameters for InternVL-Spatial-8B models.** Key configurations for InternVL-Spatial-8B, including model architectures and training parameters.

| | InternVL-Spatial-8B | |
|---|---|---|
| Model | ViT | InternViT-300M |
| | LLM | Internlm2_5-7b-chat |
| Training Hyperparameters | Tile Resolution | 448 |
| | Lora Rank | 16 |
| | Packed Batch Size | 64 |
| | Optimizer | AdamW |
| | Learning Rate | 2.00E-05 |
| | Warmup Ratio | 0.03 |
| | LR Scheduler | Cosine |
| | Weight Decay | 0.05 |
| | ViT Drop Path | 0.1 |
| | Image Tile Threshold | 40 |
| | Context Length | 12.8K |
| | Epochs | 1 |

# F  VISUALIZATION OF INTERNSPATIAL

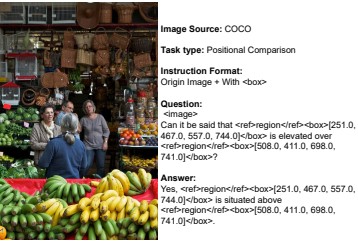

**Image Source:** COCO

**Task type:** Positional Comparison

**Instruction Format:**
Origin Image + With <box>

**Question:**
<image>
Can it be said that <ref>region</ref><box>[251.0, 467.0, 557.0, 744.0]</box> is elevated over <ref>region</ref><box>[508.0, 411.0, 698.0, 741.0]</box>?

**Answer:**
Yes, <ref>region</ref><box>[251.0, 467.0, 557.0, 744.0]</box> is situated above <ref>region</ref><box>[508.0, 411.0, 698.0, 741.0]</box>.

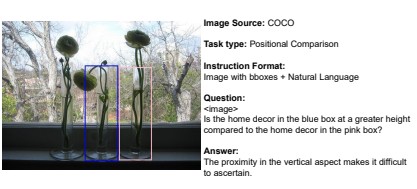

**Image Source:** COCO

**Task type:** Positional Comparison

**Instruction Format:**
Image with bboxes + Natural Language

**Question:**
<image>
Is the home decor in the blue box at a greater height compared to the home decor in the pink box?

**Answer:**
The proximity in the vertical aspect makes it difficult to ascertain.

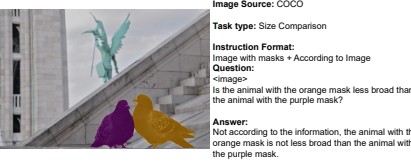

**Image Source:** COCO

**Task type:** Size Comparison

**Instruction Format:**
Image with masks + According to Image

**Question:**
<image>
Is the animal with the orange mask less broad than the animal with the purple mask?

**Answer:**
Not according to the information, the animal with the orange mask is not less broad than the animal with the purple mask.

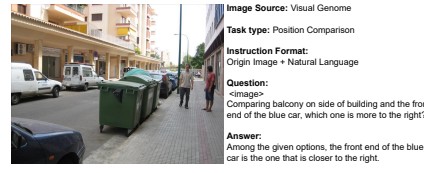

**Image Source:** Visual Genome

**Task type:** Position Comparison

**Instruction Format:**
Origin Image + Natural Language

**Question:**
<image>
Comparing balcony on side of building and the front end of the blue car, which one is more to the right?

**Answer:**
Among the given options, the front end of the blue car is the one that is closer to the right.

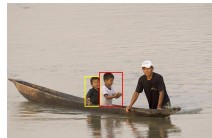

**Image Source:** Visual Genome

**Task type:** Position Comparison

**Instruction Format:**
Image with bboxes + With <ref><box>

**Question:**
<image>
Comparing <ref>region</ref><box>[367.0, 520.0, 441.0, 735.0]</box> and <ref>region</ref><box>[436.0, 490.0, 545.0, 734.0]</box>, which is more behind?

**Answer:**
<ref>region</ref><box>[367.0, 520.0, 441.0, 735.0]</box> is clearly more behind than the other.

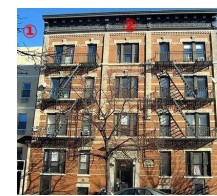

**Image Source:** Visual Genome

**Task type:** Position Comparison

**Instruction Format:**
Image with numbers + With <ref><box>

**Question:**
<image>
Between <ref>blue and clear sky</ref><box>[0.0, 3.0, 125.0, 205.0]</box> and <ref>decorative black roof on the building</ref><box>[125.0, 5.0, 977.0, 145.0]</box>, which object is closest to the observer?

**Answer:**
<ref>decorative black roof on the building</ref><box>[125.0, 5.0, 977.0, 145.0]</box> is in front.

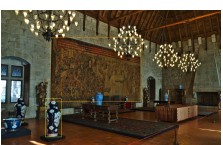

**Image Source:** AS-1B

**Task type:** Size Comparison

**Instruction Format:**
Image with bboxes+ With <ref><box>

**Question:**
<image>
Which has a greater height, the vase in the yellow box or the tapestry in the orange box?

**Answer:**
The height of the tapestry in the orange box surpasses the other.

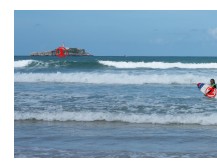

**Image Source:** AS-1B

**Task type:** Size Comparison

**Instruction Format:**
Image with numbers+ According to Image

**Question:**
<image>
Which has a greater width, the island with a circled '1' drawn on it or the surfer with a circled '2' drawn on it?

**Answer:**
The width of the island with a circled '1' drawn on it is greater.

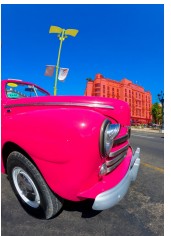

**Image Source:** AS-1B

**Task type:** Size Comparison

**Instruction Format:**
Image with masks+ With <box>

**Question:**
<image>
Is <ref>region</ref><box>[295.0, 89.0, 475.0, 374.0]</box> larger than <ref>region</ref><box>[510.0, 282.0, 983.0, 495.0]</box>?

**Answer:**
Sorry, but <ref>region</ref><box>[510.0, 282.0, 983.0, 495.0]</box> is bigger than <ref>region</ref><box>[295.0, 89.0, 475.0, 374.0]</box>.

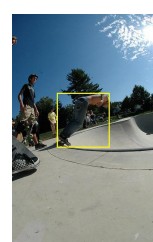

**Image Source:** COCO

**Task type:** Existence Estimation

**Instruction Format:**
Origin Image + With <box>

**Question:**
<image>
Are there any people closer to the observer than <ref>region</ref><box>[282.0, 331.0, 615.0, 558.0]</box>?

**Answer:**
No.

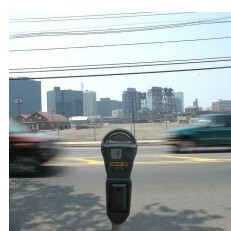

**Image Source:** COCO

**Task type:** Object Counting

**Instruction Format:**
Origin Image + Natural Language

**Question:**
<image>
How many cars are positioned lower than the small house with a red roof?

**Answer:**
2.

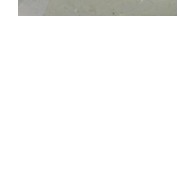

**Image Source:** Cityscapes

**Task type:** Size Comparison

**Instruction Format:**
Image with numbers+ According to Image

**Question:**
<image>
the car with a circled '1' drawn on it is smaller than the car with a circled '2' drawn on it, isn't it?

**Answer:**
Definitely, the car with a circled '1' drawn on it is smaller than the car with a circled '2' drawn on it.

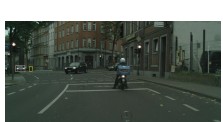

**Image Source:** Cityscapes

**Task type:** Size Comparison

**Instruction Format:**
Image with bboxes + According to Image

**Question:**
<image>
Which is larger, the bicycle in the yellow box or the car in the pink box?

**Answer:**
the car in the pink box is the larger one.

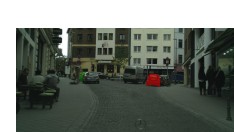

**Image Source:** Cityscapes

**Task type:** Positional Comparison

**Instruction Format:**
Image with masks + According to Image

**Question:**
<image>
Can you determine if the car with the red mask is closer or farther from the observer compared to the car with the green mask?

**Answer:**
the car with the red mask is in front of the car with the green mask.

## G VISUALIZATION OF RESULTS ON INTERNSPATIAL-BENCH

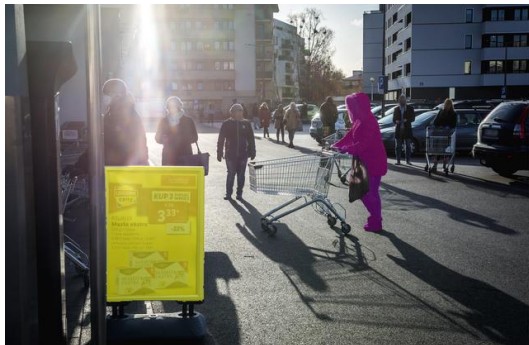

Question: <image> Where is <ref>the woman in the red coat</ref> positioned in relation to <ref>the sign</ref>? Answer with the option's letter from the given choices directly. (A) To the right side of <ref>the sign</ref> (B) In front of <ref>the sign</ref> (C) Behind <ref>the sign</ref> (D) To the left side of <ref>the sign</ref>

GT: A

InternVL-8B Pred: (C) Behind the sign

InternVL-Spatial-8B Pred: (A) To the right side of the sign

Task: positional

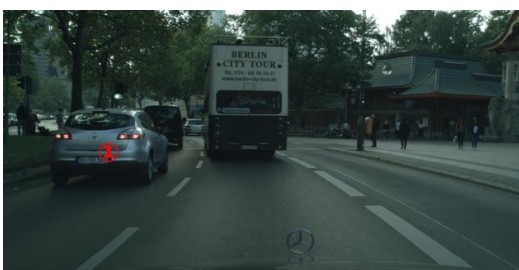

Question: <image> Are there any vehicles more to the right than <ref>the silver car in the left lane</ref>? Answer with a single word or option's letter.

GT: Yes

InternVL-8B Pred: No

InternVL-Spatial-8B Pred: Yes

Task: existence

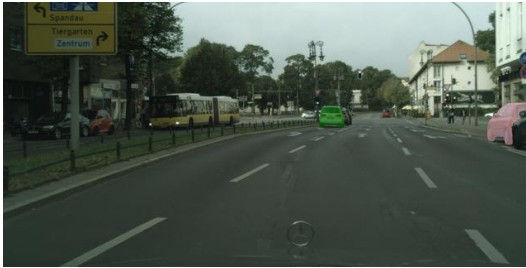

Question: <image> Is <ref>region</ref><box>[[599, 390, 654, 472]]</box> positioned farther from the observer than <ref>region</ref><box>[[918, 378, 1000, 538]]</box>? Answer with a single word or option's letter.

GT: Yes

InternVL-8B Pred: No

InternVL-Spatial-8B Pred: Yes

Task: positional

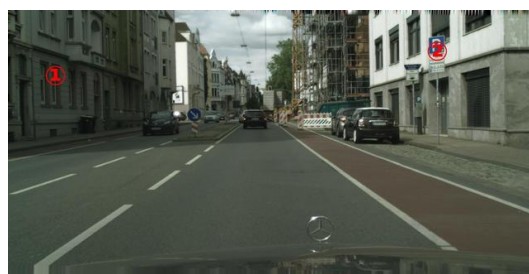

Question: <image> Is <ref>the blue parking sign on the right side of the road</ref> shorter than <ref>the white building on the left side</ref>? Answer with a single word or option's letter.

GT: Yes

InternVL-8B Pred: No

InternVL-Spatial-8B Pred: Yes

Task: size

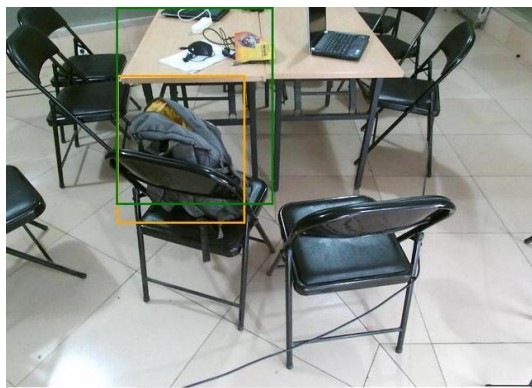

Question: <image> Which is above, <ref>region</ref><box>[210, 179, 453, 566]</box> or <ref>region</ref><box>[209, 0, 506, 516]</box>?

GT: Positioned higher is <ref>region</ref><box>[209, 0, 506, 516]</box>.

InternVL-8B Pred: region[210, 179, 453, 566] is above region[209, 0, 506, 516].

InternVL-Spatial-8B Pred: region[209, 0, 506, 516] is the one that is above.

Task: above_choice

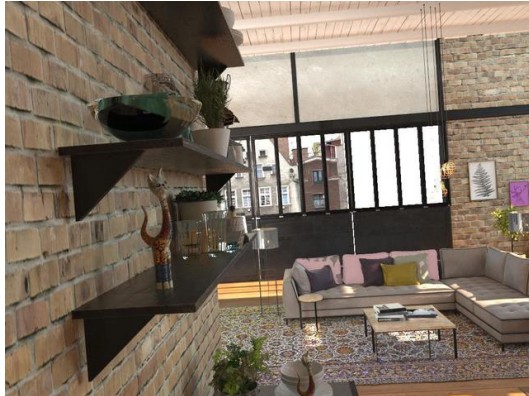

Question: <image> Is the picture with the purple mask below the pillow with the pink mask?

GT: Actually, the picture with the purple mask is above the pillow with the pink mask.

InternVL-8B Pred: Yes, the picture with the purple mask is below the pillow with the pink mask.

InternVL-Spatial-8B Pred: No, the picture with the purple mask is not situated beneath the pillow with the pink mask.

Task: below_predicate

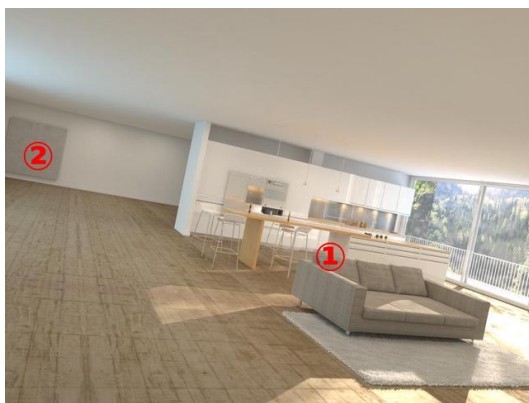

Question: <image> Does the counter with a circled '1' drawn on it come behind the picture with a circled '2' drawn on it?

GT: No, it is not.

InternVL-8B Pred: Yes, the counter with a circled '1' is behind the picture with a circled '2'.

InternVL-Spatial-8B Pred: No, the counter with a circled '1' drawn on it is not behind the picture with a circled '2' drawn on it.

Task: behind_predicate

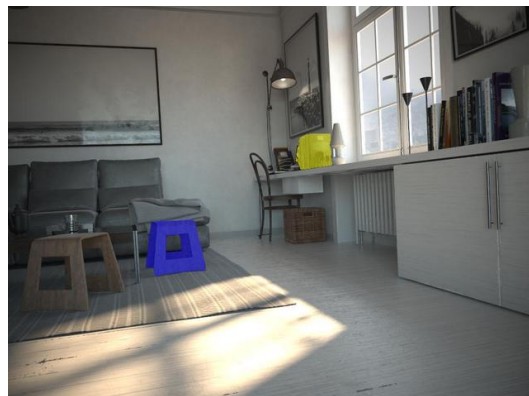

Question: <image> Does the books with the yellow mask come in front of the chair with the blue mask?

GT: No, it is not.

InternVL-8B Pred: Yes, the books with the yellow mask are placed in front of the chair with the blue mask.

InternVL-Spatial-8B Pred: No, the books with the yellow mask is not in front of the chair with the blue mask.

Task: front_predicate

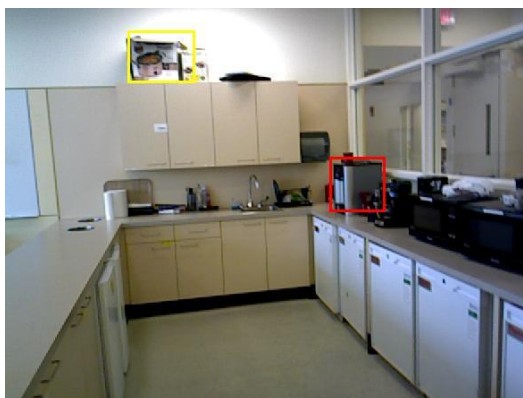

Question: <image> Which is more to the left, the coffee maker in the red box or the box in the yellow box?

GT: From the viewer's perspective, the box in the yellow box appears more on the left side.

InternVL-8B Pred: The coffee maker in the red box is more to the left than the box in the yellow box.

InternVL-Spatial-8B Pred: the box in the yellow box is positioned further to the left.

Task: left_choice

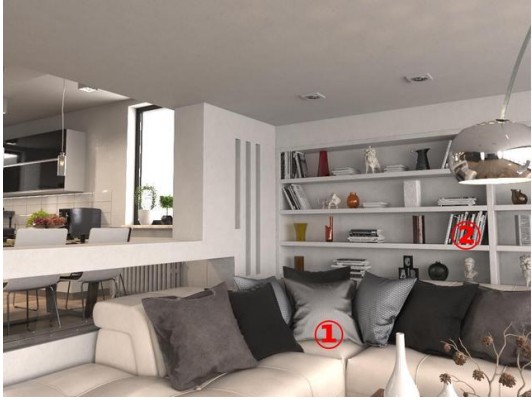

Question: <image> Can you confirm if the pillow with a circled '1' drawn on it is positioned to the right of the books with a circled '2' drawn on it?

GT: No, the pillow with a circled '1' drawn on it is not to the right of the books with a circled '2' drawn on it.

InternVL-8B Pred: Yes, the pillow with a circled '1' is positioned to the right of the books with a circled '2'.

InternVL-Spatial-8B Pred: the pillow with a circled '1' drawn on it is not to the right of the books with a circled '2' drawn on it, actually.

Task: right_predicate

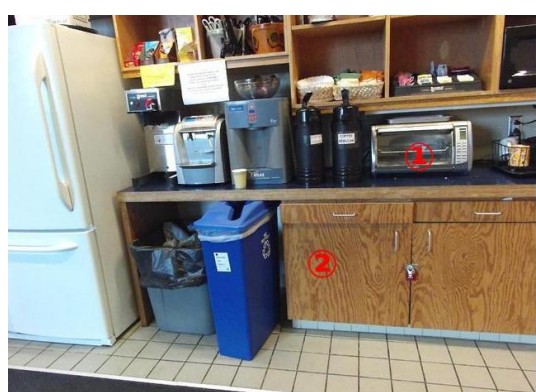

Question: <image> Does the toaster with a circled '1' drawn on it have a larger size compared to the counter with a circled '2' drawn on it?

GT: Incorrect, the toaster with a circled '1' drawn on it is not larger than the counter with a circled '2' drawn on it.

InternVL-8B Pred: Yes, the toaster with a circled '1' drawn on it is larger in size compared to the counter with a circled '2' drawn on it.

InternVL-Spatial-8B Pred: Not according to the information, the toaster with a circled '1' drawn on it is not larger than the counter with a circled '2' drawn on it.

Task: big_predicate

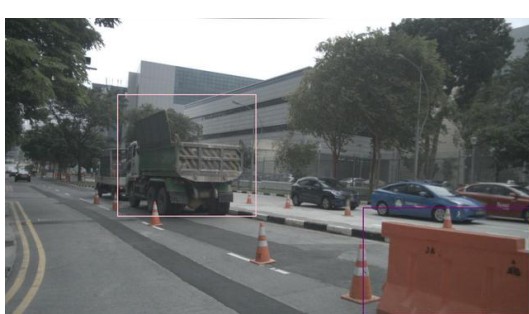

Question: <image> Does the barrier in the purple box have a greater height compared to the truck in the pink box?

GT: In fact, the barrier in the purple box may be shorter than the truck in the pink box.

InternVL-8B Pred: Yes, the barrier in the purple box appears to have a greater height compared to the truck in the pink box.

InternVL-Spatial-8B Pred: Sorry, but the barrier in the purple box does not exceed the truck in the pink box in height.

Task: tall_predicate

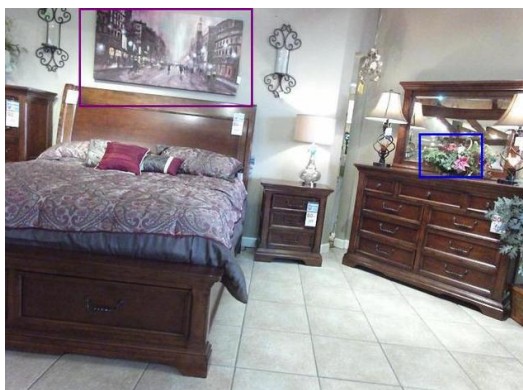

Question: <image> Can you confirm if the painting in the purple box is thinner than the potted plant in the blue box?

GT: In fact, the painting in the purple box might be wider than the potted plant in the blue box.

InternVL-8B Pred: Yes, the painting in the purple box is thinner than the potted plant in the blue box.

InternVL-Spatial-8B Pred: No, the painting in the purple box is not narrower than the potted plant in the blue box.

Task: thin_predicate

## H   THE USE OF LARGE LANGUAGE MODELS (LLMS)

We used large language models (LLMs) as assistive tools during the preparation of this work. Specifically, LLMs were employed for language polishing, LaTeX code editing, and debugging of prompts in the dataset construction process. The authors take full responsibility for the content of the paper.

