# OpenReview forum: "InternSpatial: A Comprehensive Dataset for Spatial Reasoning in Vision-Language Models"
_ICLR.cc/2026/Conference — ICLR 2026 Poster_

### Official Review · Reviewer_mDLs · 2025-10-30

**Soundness:** 2
**Presentation:** 2
**Contribution:** 2
**Rating:** 4
**Confidence:** 4

**Summary:**

Authors introduce a 12M QA-pair dataset containing both single and multi view images. This dataset aims to improve spatial reasoning in vision-language models (VLMs). The data is sourced from diverse domains (in-the-wild, indoor, driving, object-centric, embodied) and also contains an evaluation benchmark of 6,000 QAs.
The data generation pipeline uses automated annotation (e.g. object masks, depth estimation) with pretrained model followed by template-based QA generation.
The authors finetune an InternVL2.5-8B models on their dataset and obtain clear performance gains on spatial reasoning benchmarks including their own, while maintaining performance on general VQA benchmarks.

**Strengths:**

1. Clear improvement over baseline when trained on new data.
2. Thorough analysis of dataset statistics.
3. Large-scale open-source dataset contribution

**Weaknesses:**

1. Only `InternVL2.5-8B` is used as a baseline. How does this dataset help other models in general?
2. Only an 8B scale model is used. Will this data improve smaller (e.g. 2B-3B models), large (14B-70B), and MoE models?
3. Minimal description of data generation process. How do the automated tools (e.g. SAM for bbox / mask) perform? What are the error rates? These are not clearly discussed in the paper.
4. QA generation: diversity? These is little analysis on the dataset. Maybe calculate diversity metrics on the text and image data. Also, given the template based generation (as opposed to human or LLM), this data diversity concern is amplified.
5. Test data leakage: The training dataset is created using 3D information of some datasets, "integrated multi-view data derived from the training splits of the ScanNet/MultiScan/R2R/Objaverse". At the same time, their benchmark uses some of these same datasets' test splits to evaluate. Even other benchmark (e.g. VSI contains ScanNet data) use this common data. Could the performance improvement be due to this similar domain data being used to create the training dataset?

**Questions:**

See weaknesses.

1) In Table 1, the authors mention their dataset "InternSpatial" as Open-source. Is the dataset already open-source? If not, this claim is false?

2) See related work sections of below papers for 2D spatial reasoning. Consider discussing prior 2D spatial reasoning works in detail?
  - Ferretv2: https://arxiv.org/abs/2404.07973
  - LocVLM: https://arxiv.org/abs/2404.07449
  - Shikra: https://arxiv.org/pdf/2306.15195

---

> ### Author Response · Authors · 2025-11-25
>
> We would like to sincerely thank all reviewers for their constructive comments and valuable suggestions. Their insightful feedback has helped us identify opportunities to further clarify our methodology and strengthen the experimental analysis. We carefully address each question point-by-point below.
>
> ## Only InternVL2.5-8B is used as a baseline. How does this dataset help other models in general? && Only an 8B scale model is used. Will this data improve smaller (e.g. 2B-3B models), large (14B-70B), and MoE models?
> To evaluate whether the proposed dataset benefits architectures beyond InternVL2.5-8B, we additionally fine-tuned **Qwen2.5-VL-8B** and **InternVL2.5-1B** using **InternSpatial**. The detailed results are provided in the revised Table. Similar to the InternVL2.5-8B experiments, the **Qwen2.5-VL-8B** model and **InternVL2.5-1B** trained with **InternSpatial** achieve consistent improvements on both **InternSpatial-Bench** and **VSI-Bench**, demonstrating that the gains are not tied to a specific model architecture or a specific model size.These results indicate that our dataset provides generalizable spatial supervision that transfers across different open-source VLM frameworks and different model size.
>
> **Results of Qwen2.5-VL-8B, InternVL2.5-1B and the corresponding finetuned model (Qwen-Spatial-8B, InternVL-Spatial-1B) on InternSpatial-Bench.**
> |Model|Position Comparison|Size Comparison|Rotation Estimation|Object Counting|Existence Estimation|Average|
> |:-----:|:-----:|:-----:|:-----:|:-----:|:-----:|:-----:|
> |Qwen2.5-VL-8B|57.1|60.8|26.9|58.0|66.7|53.9|
> |Qwen-Spatial-8B|79.9(+22.8)|78.7(+17.9)|34.4(+7.5)|68.3(+10.3)|80(+13.3)|68.3(+14.4)|
> |InternVL2.5-1B|42.9|43.3|23.8|21.3|59.9|38.2|
> |InternVL-Spatial-1B|65.4(+22.5)|58.5(+15.2)|26.3(+2.5)|59.4(+28.1)|74.4(+14.5)|56.8(+18.6)|
>
> **Results of Qwen2.5-VL-8B, InternVL2.5-1B and the corresponding finetuned model (Qwen-Spatial-8B, InternVL-Spatial-1B) on VSI-Bench.**
> |Model|Obj.Count|Abs.Dist.|Obj.size|Room Size|Rel.Dist.|Route Plan|Appr.Order|Average|
> |:-----:|:-----:|:-----:|:-----:|:-----:|:-----:|:-----:|:-----:|:-----:|
> |Qwen2.5-VL-8B|41.5|21.2|50.7|36.6|37.9|30.4|34.0|36.0|
> |Qwen-Spatial-8B|60.8(+19.3)|35.0(+13.8)|53.4(+2.7)|45.0(+8.4)|40.0(+2.1)|36.6(+6.2)|34.5(+0.5)|43.6(+7.6)|
> |InternVL2.5-1B|51.8|3.9|24.8|13.7|25.6|32.5|7.6|22.8|
> |InternVL-Spatial-1B|66.4(14.6)|25.4(+21.5)|42.0(+17.2)|48.5(+24.8)|34.1(+8.5)|34.0(+1.5)|11.0(+3.4)|37.3(+14.5)|
>
> ## Minimal description of data generation process. How do the automated tools (e.g. SAM for bbox / mask) perform? What are the error rates? These are not clearly discussed in the paper.
> We conducted manual verification on a randomly sampled subset, including both final QA and intermediate results. The results showed that the accuracy on the subset was over 90%. Most errors were concentrated in counting tasks for single images, as this requires the model to accurately identify all objects of a specified category in the first step of the pipeline. In some images with a large number of objects of the same type or with objects that are difficult to distinguish, accurately identifying all objects is a significant challenge for current VLM models. We will improve the analysis and discussion of dataset quality and workflow accuracy in future revisions.
>
> ## QA generation: diversity? These is little analysis on the dataset. Maybe calculate diversity metrics on the text and image data. Also, given the template based generation (as opposed to human or LLM), this data diversity concern is amplified.
> We used a wide range of data sources to construct our dataset, including in-the-wild, indoor, urban, and single-object scenes. We also considered a rich set of instruction formats.  The distribution of our dataset was analyzed statistically in Figure 3 and 4 in Section 3.3.
>
> We construct QAs through a LLM-designed template library to keep a balance between linguistic diversity and efficiency of our pipeline. In detail, we prompt LLM to generate several (typically 3-4) templates with different sentence structures for each type of questions and answers. These templates were randomly chosen when building QAs in **InternSpatial** dataset. To ensure the quality of the template library, we further performed a human check on all templates. Here shows an example:
> ```
> far_choice_templates = {
>     "question_templates": [
>        "Which one is further back, [A] or [B]?",
>        "Can you tell me which is positioned more towards the back, [A] or [B]?",
>        "Between [A] and [B], which is more distant in the rear aspect?",
>        "Comparing [A] and [B], which is more behind?"
>     ],
>     "answer_templates": [
>        "[O] is definitely more behind.",
>        "I can confirm that [O] is situated further back.",
>        "[O] is clearly more behind than the other.",
>        "There is no question that [O] is more behind."
>     ]
> }
> ```
> The full content of our template library is shown in **Appendix B**.

---

> ### Author Response · Authors · 2025-11-25
>
> ## Test data leakage
> Regarding test data leakage, all training data in our dataset are generated strictly from the training splits of ScanNet, MultiScan, R2R, and Objaverse. We carefully verify every source to ensure that no test images, frames, or 3D reconstructions from these datasets are used in our pipeline, thereby eliminating the possibility of direct test leakage.
>
> As for whether the performance gains could be partly attributed to domain similarity, we acknowledge that this is a reasonable concern. Many video spatial-reasoning benchmarks, including **VSI-Bench**, are built on indoor scenes, which are also the dominant domain in widely used 3D datasets. Because existing benchmarks largely share this domain, it is challenging to isolate the exact contribution of domain similarity without dedicated cross-domain experiments. We agree that evaluating out-of-domain generalization (e.g., outdoor or non-indoor 3D scenes) is an important direction, and we plan to explore this in future work.
>
> ## In Table 1, the authors mention their dataset "InternSpatial" as Open-source. Is the dataset already open-source? If not, this claim is false?
> We will release both the data generation pipeline and the training data in the near future.
>
> ## See related work sections of below papers for 2D spatial reasoning. Consider discussing prior 2D spatial reasoning works in detail?
> We sincerely acknowledge that the current manuscript does not sufficiently discuss prior work on 2D spatial reasoning.
>
> Ferret-v2 [1] significantly enhances fine-grained regional referring and grounding, enabling models to more accurately identify and localize arbitrary-shaped regions within an image. LocVLM [2] injects spatial awareness via instruction fine-tuning with image-space coordinates, leading to markedly better spatial perception and object localization (e.g., resolving common weaknesses such as left–right confusion) and improving VQA performance while reducing hallucinations. Shikra [3] further introduces a paradigm that represents spatial coordinates purely in natural language, unifying position inputs and outputs within a single architecture and enabling a multimodal LLM to handle both location-annotated inputs and region-referring outputs in referential dialogues.
>
> However, these approaches also have limitations. The Ferret family focuses primarily on region-level referring and grounding and remains constrained by the resolution and design of the underlying pre-trained visual encoder. LocVLM, while effective at enhancing spatial cognition, relies on a relatively limited amount of pseudo-labeled data and does not cover a broad spectrum of spatial reasoning task forms. Shikra combines several existing datasets with GPT-generated annotations, but its overall data scale and instruction diversity are still limited, and it does not address more complex settings such as multi-view spatial reasoning. In contrast, InternSpatial is explicitly designed to fill these gaps: it is, to our knowledge, the largest open-source spatial reasoning dataset (12M QA pairs) spanning both single-view and multi-view scenarios, with rich visual diversity and 19 distinct instruction formats that support fine-grained spatial relation understanding and a wide variety of query styles. Moreover, we introduce novel tasks such as multi-view rotation-angle prediction, an aspect that, to the best of our knowledge, has not been explored in prior spatial VLM work, which substantially broadens the task coverage from local position relations to cross-view spatial cognition. Empirically, models trained on InternSpatial achieve double-digit improvements on both InternSpatial-Bench and existing benchmarks such as VSI-Bench, while maintaining strong performance on general-purpose tasks, demonstrating that our dataset provides a strong foundation for advancing spatially capable VLMs.
>
> Taken together, our work complements and extends Ferret-v2, LocVLM, and Shikra by offering higher-precision spatial understanding, a more comprehensive data design, and substantially broader task coverage. We will add a dedicated paragraph in the revised manuscript to discuss these works and clearly position InternSpatial with respect to them.
>
> [1] ZHANG, Haotian, et al. Ferret-v2: An improved baseline for referring and grounding with large language models. arXiv preprint arXiv:2404.07973, 2024.
>
> [2] RANASINGHE, Kanchana, et al. Learning to localize objects improves spatial reasoning in visual-llms. In: Proceedings of the IEEE/CVF Conference on Computer Vision and Pattern Recognition. 2024. p. 12977-12987.
>
> [3] CHEN, Keqin, et al. Shikra: Unleashing multimodal llm's referential dialogue magic. arXiv preprint arXiv:2306.15195, 2023.
>
> We appreciate the reviewers’ thoughtful evaluation of our work. We have incorporated the additional clarifications, analyses, and results discussed in this rebuttal into the revised manuscript.

---

### Official Review · Reviewer_2hUS · 2025-11-02

**Soundness:** 2
**Presentation:** 2
**Contribution:** 3
**Rating:** 4
**Confidence:** 3

**Summary:**

This paper introduces InternSpatial and the corresponding benchmark for spatial reasoning. The dataset contains 12 million QA pairs spanning both single-view and multi-view scenarios, covering 19 instruction formats (textual and visual). It also proposes InternSpatial-Bench, a new benchmark with a new rotation angle prediction task. Experiments are performed on various benchmarks.

**Strengths:**

1.	The proposed dataset is large-scale especially regarding the number of QA pairs.
2.	The data generation pipeline is sound.
3.	It shows promising performance leveraging the curated data.

**Weaknesses:**

1.	Could the author compare the proposed dataset with previous ones in terms of scenarios, question types, etc.?
2.	Could the authors validate the effectiveness of the proposed data on more open-source frameworks?
3.	Could the authors explain the performance show limited gain on rotation estimation and object counting?
4.	Could the generated QA pairs reflect the complexity or ambiguity of human spatial questions.
5.	Will data generation pipeline and data be publicly available?

**Questions:**

Please refer to Weakness.

---

> ### Author Response · Authors · 2025-11-25
>
> We would like to sincerely thank all reviewers for their constructive comments and valuable suggestions. Their insightful feedback has helped us identify opportunities to further clarify our methodology and strengthen the experimental analysis. We carefully address each question point-by-point below.
>
> ## Could the author compare the proposed dataset with previous ones in terms of scenarios, question types, etc.?
> As stated in the Introduction, **InternSpatial** includes both single-view and multi-view samples across diverse scenes, and supports **19 instruction formats** that cover a wide range of spatial query types. A detailed comparison with previous datasets—covering aspects such as scenarios, question types, and annotation formats—is provided in **Table 1** of the paper:
>
> **Table 1: Comparison of our InternSpatial with existing spatial reasoning datasets. W: in-the-wild, I: indoor, D: drive, E: embodied, O: object-centric**
> | Dataset                             | # of QA | Scenario     | Open-source | View Type                  | Instruction format   |
> |:-------------------------------------:|:---------:|:--------------:|:-------------:|:----------------------------:|:----------------------:|
> | SpatialVLM (Chen et al., 2024)      | 2B      | W            | ✗           | Single-view                 | Single-format       |
> | SpatialQA (Cai et al., 2025)        | 0.9M    | W, E         | ✓           | Single-view                 | Single-format       |
> | OSD (Cheng et al., 2024)            | 8.7M    | W            | ✓           | Single-view                 | Single-format       |
> | InternSpatial                       | 12M     | W, I, D, E, O| ✓           | Single-view, Multi-view     | Multiple-format     |
>
> ## Could the authors validate the effectiveness of the proposed data on more open-source frameworks?
> To evaluate whether the proposed dataset benefits architectures beyond InternVL2.5-8B, we additionally fine-tuned **Qwen2.5-VL-8B** using **InternSpatial**. The detailed results are provided in the revised Table. Similar to the InternVL2.5-8B experiments, the Qwen2.5-VL-8B model trained with **InternSpatial** achieves consistent improvements on both **InternSpatial-Bench** and **VSI-Bench**, demonstrating that the gains are not tied to a specific model architecture.These results indicate that our dataset provides generalizable spatial supervision that transfers across different open-source VLM frameworks.
>
> **Results of Qwen2.5-VL-8B and the corresponding finetuned model (Qwen-Spatial-8B) on InternSpatial-Bench.**
> |Model|Position Comparison|Size Comparison|Rotation Estimation|Object Counting|Existence Estimation|Average|
> |:-----:|:-----:|:-----:|:-----:|:-----:|:-----:|:-----:|
> |Qwen2.5-VL-8B|57.1|60.8|26.9|58.0|66.7|53.9|
> |Qwen-Spatial-8B|79.9(+22.8)|78.7(+17.9)|34.4(+7.5)|68.3(+10.3)|80(+13.3)|68.3(+14.4)|
>
> **Results of Qwen2.5-VL-8B and the corresponding finetuned model (Qwen-Spatial-8B) on VSI-Bench.**
> |Model|Obj.Count|Abs.Dist.|Obj.size|Room Size|Rel.Dist.|Route Plan|Appr.Order|Average|
> |:-----:|:-----:|:-----:|:-----:|:-----:|:-----:|:-----:|:-----:|:-----:|
> |Qwen2.5-VL-8B|41.5|21.2|50.7|36.6|37.9|30.4|34.0|36.0|
> |Qwen-Spatial-8B|60.8(+19.3)|35.0(+13.8)|53.4(+2.7)|45.0(+8.4)|40.0(+2.1)|36.6(+6.2)|34.5(+0.5)|43.6(+7.6)|
>
> ## Could the authors explain the performance show limited gain on rotation estimation and object counting?
> For **Rotation Estimation**, the task is inherently challenging: even strong VLMs such as Qwen2.5-VL achieve only **30.6%** accuracy. Our improvement from **28.5% to 33.6%** likely reflects the upper bound attainable by an 8B-scale model, suggesting that model capacity—not data alone—may be the limiting factor.
> For **Object Counting**, the baseline InternVL-8B model already exhibits strong counting capability, outperforming most existing VLMs and achieving competitive results even before incorporating our spatial training. As a result, the available performance headroom is relatively small, leading to a more modest improvement after training InternVL-Spatial-8B.

---

> > ### Author Response · Authors · 2025-11-25
> >
> > ## Could the generated QA pairs reflect the complexity or ambiguity of human spatial questions.
> > We ensure the language complexity through a LLM-designed template library. In detail, we prompt LLM to generate several (typically 3-4) templates with different sentence structures for each type of questions and answers. These templates were randomly chosen when building QAs in **InternSpatial** dataset. Here is an example of our templates:
> > ```
> > near_predict_templates = {
> >     "question_templates": [
> >        "Is [A] positioned in front of [B]?",
> >        "Does [A] precede [B] in this arrangement?",
> >        "Is [A] in front of [B]?",
> >        "Is [A] closer to the observer than [B]?"
> >     ],
> >     "positive_answer_templates": [
> >        "Without a doubt, [A] stands nearer to the viewer than [B].",
> >        "Definitely, [A] is more proximate to the observer than [B].",
> >        "Indeed, [A] is in front of [B].",
> >        "Absolutely, [A] is before [B].",
> >     ],
> >     "negative_answer_templates": [
> >        "Not at all, [A] is not closer to the observer than [B].",
> >        "No, [A] is not in front of [B].",
> >        "Unfortunately, [A] is not ahead of [B].",
> >        "Definitely not, [A] is not closer to the observer than [B]."
> >     ]
> > }
> > ```
> > **Appendix B** contains the detail of our template library.
> >
> > The ambiguity of questions should be avoided in the trainset and the benchmark. For this purpose, we involved several filter strategies in our pipeline. For in-the-wild images, we filtered out objects without clear boundaries by object category, such as the sky and grass. For indoor scenes with 3D annotations, we detected the occlusions and objects beyond the image boundary through the projection of 3D models and bboxes on the image plane, and excluded QA pairs that were ambiguous due to these situations.
> >
> > ## Will data generation pipeline and data be publicly available?
> > Yes. We will release both the data generation pipeline and the training data in the near future.
> >
> > We appreciate the reviewers’ thoughtful evaluation of our work. We have incorporated the additional clarifications, analyses, and results discussed in this rebuttal into the revised manuscript.

---

### Official Review · Reviewer_S4PX · 2025-11-03

**Soundness:** 3
**Presentation:** 3
**Contribution:** 3
**Rating:** 6
**Confidence:** 3

**Summary:**

This paper proposes InternSpatial, which is claimed to be the largest spatial QA dataset with 12M data. They sourced the data from 2D images as well as 3D datasets of various sources, generating single-view and multi-view QA pairs. They report the scores of various models on the InternSpatial Bench, including InternVL-8B model trained on their datasets. They also report results on VSI-Bench as well as other general benchmarks.

**Strengths:**

In general, I think a12M dataset is a quite significant improvement from previous QA datasets in terms of data quantity. The dataset comes from a wide variety of data, as shown by Figure 4. Results on InternSpatial-Bench, VSI-Bench as well as other general benchmark results show that training on this dataset brings a lot of improvements.

**Weaknesses:**

It would be great to understand how much the image datasets are helping with the training in general. The alignment to view space from 2D images requires depth estimation followed by camera estimation, both of which could potentially introduce significant errors. I wonder if it would be possible to see the improvements based on InternVL-Spatial-8B trained with only 3D datasets and/or only 2D datasets. Also, this would give more insights on whether there are domain gaps within the training dataset itself.

The paper could also be strengthened by showing more baselines of models specialized in 3D on InternSpatial-Bench (e.g., SpatialMLLM, SpaceR, etc). Currently, the results shown are mainly on general VLMs and not VLMs specifically trained for spatial reasoning.

The paper would also be improved by showing results of more methods finetuned with InternSpatial dataset to further show the effectiveness of the dataset on different model architectures/training methods.

**Questions:**

Overall, my main questions of the paper are two-fold:

1. Are both dataset sources (2D and 3D images) helpful in contributing to the training when evaluating on other benchmarks such as VSI-Bench? Are there potential ways to filter out results of bad prediction during 2D data preprocessing?

2.Are there results of other methods finetuned on this dataset? Does that bring any further improvements?

Overall, I do see this dataset being beneficial to the community, so I am leaning towards accept. The questions I believe would significantly add to the contribution.

---

> ### Author Response · Authors · 2025-11-25
>
> We would like to sincerely thank all reviewers for their constructive comments and valuable suggestions. Their insightful feedback has helped us identify opportunities to further clarify our methodology and strengthen the experimental analysis. We carefully address each question point-by-point below.
>
> ## Are both dataset sources (2D and 3D images) helpful in contributing to the training when evaluating on other benchmarks such as VSI-Bench?
> Because the task types in our 2D and 3D image datasets differ, training solely on 2D data does not provide improvements across all tasks in **VSI-Bench**, which primarily focuses on 3D spatial understanding. Nevertheless, we do observe gains on tasks that are more closely aligned with 2D spatial reasoning. For example, training only on 2D images improves **Object Counting** (from 51.7% to 54.8%), **Relative Distance Estimation** (from 40.8% to 41.7%), and **Route Planning** (from 27.8% to 34.5%). This suggests that while 2D data cannot fully substitute for 3D training signals, it still contributes positively to tasks with shared spatial characteristics.
>
> ## Are there potential ways to filter out results of bad prediction during 2D data preprocessing?
> We added several filtering strategies to our data pipeline. For in-the-wild images, we filtered out objects without clear boundaries by object category, such as the sky and grass. For indoor scenes with 3D annotations, we detected the occlusions and objects beyond the image boundary through the projection of 3D models and bboxes on the image plane, and excluded QA pairs that were ambiguous due to these situations. In this way, we aim to reduce the interference caused by ambiguous or unanswerable QA pairs.
>
> ## Are there results of other methods finetuned on this dataset? Does that bring any further improvements?
> To evaluate whether the proposed dataset benefits architectures beyond InternVL2.5-8B, we additionally fine-tuned **Qwen2.5-VL-8B** using **InternSpatial**. The detailed results are provided in the revised Table. Similar to the InternVL2.5-8B experiments, the Qwen2.5-VL-8B model trained with InternSpatial achieves consistent improvements on both **InternSpatial-Bench** and **VSI-Bench**, demonstrating that the gains are not tied to a specific model architecture.These results indicate that our dataset provides generalizable spatial supervision that transfers across different open-source VLM frameworks.
>
> **Results of Qwen2.5-VL-8B and the corresponding finetuned model (Qwen-Spatial-8B) on InternSpatial-Bench.**
> |Model|Position Comparison|Size Comparison|Rotation Estimation|Object Counting|Existence Estimation|Average|
> |:-----:|:-----:|:-----:|:-----:|:-----:|:-----:|:-----:|
> |Qwen2.5-VL-8B|57.1|60.8|26.9|58.0|66.7|53.9|
> |Qwen-Spatial-8B|79.9(+22.8)|78.7(+17.9)|34.4(+7.5)|68.3(+10.3)|80(+13.3)|68.3(+14.4)|
>
> **Results of Qwen2.5-VL-8B and the corresponding finetuned model (Qwen-Spatial-8B) on VSI-Bench.**
> |Model|Obj.Count|Abs.Dist.|Obj.size|Room Size|Rel.Dist.|Route Plan|Appr.Order|Average|
> |:-----:|:-----:|:-----:|:-----:|:-----:|:-----:|:-----:|:-----:|:-----:|
> |Qwen2.5-VL-8B|41.5|21.2|50.7|36.6|37.9|30.4|34.0|36.0|
> |Qwen-Spatial-8B|60.8(+19.3)|35.0(+13.8)|53.4(+2.7)|45.0(+8.4)|40.0(+2.1)|36.6(+6.2)|34.5(+0.5)|43.6(+7.6)|
>
> We appreciate the reviewers’ thoughtful evaluation of our work. We have incorporated the additional clarifications, analyses, and results discussed in this rebuttal into the revised manuscript.

---

### Official Review · Reviewer_9QLW · 2025-11-05

**Soundness:** 3
**Presentation:** 3
**Contribution:** 3
**Rating:** 8
**Confidence:** 3

**Summary:**

This paper presents InternSpatial, a large-scale open-source dataset (12M QA pairs) designed to improve spatial reasoning in Vision-Language Models (VLMs). It addresses key limitations in prior works, such as limited scene diversity, narrow instruction formats, and lack of multi-view supervision, by aggregating data from a wide range of sources (COCO, Visual Genome, ScanNet, Cityscapes, Objaverse, R2R, etc.) and generating question-answer pairs with 19 instruction formats spanning both textual and visual variations. The authors also propose InternSpatial-Bench, a benchmark comprising 6,008 QA pairs that evaluate single-view and multi-view spatial reasoning, including a new rotation angle prediction task.

Models trained on InternSpatial (notably, InternVL-Spatial-8B) show large performance gains +12.1% on InternSpatial-Bench and +10.7% on VSI-Bench, while maintaining comparable performance on general VQA tasks, confirming that spatial reasoning gains do not come at the expense of general multimodal ability

**Strengths:**

### Reasonable Dataset Design
- The dataset covers diverse visual domains (indoor, outdoor, object-centric, embodied, urban) and both single-view and multi-view reasoning setups.
- It supports a wide variety of instruction modalities, text, bounding boxes, masks, numeric indicators, coordinate-based prompts, totaling 19 instruction types, a major advancement over prior datasets like SpatialVLM or OSD.
### Data Generation Process
- The data pipeline integrates multiple pretrained modules for depth, segmentation, and camera parameter estimation (SAM2, Metric3Dv2, PerspectiveFields, WildCamera) to lift 2D annotations into 3D canonical view space. The pipeline is modular and reproducible, allowing flexible annotation generation and QA synthesis without relying on expensive LLM prompting for each sample.

### Novel Multi-view Reasoning Component
- The addition of rotation angle prediction is new and well-motivated for embodied AI and robotics. Multi-view QA construction uses geometric consistency (e.g., Alpha Shape–based room estimation, OrientedBoundingBox fitting) to ensure physically grounded reasoning.

### Strong Experimental Results and Benchmarking
- The evaluation suite is broad: InternSpatial-Bench, VSI-Bench, and five standard multimodal tasks.
- Results show large, consistent improvements in spatial reasoning, including outperforming commercial VLMs like GPT-4o and Claude 3.7 Sonnet in several spatial tasks.
- Ablation studies isolate the effects of instruction format diversity and confirm its value for cross-format generalization.

**Weaknesses:**

### The limit of Template-Driven QA Generation.

While efficient, the template-based QA generation may lead to limited linguistic diversity and potential overfitting to templated phrasing. The authors acknowledge this, but do not quantify how template rigidity affects generalization to natural human queries.

### Lack of Qualitative Error Analysis

The evaluation focuses almost exclusively on quantitative metrics. There is little qualitative examination of failure modes (e.g., reasoning about occluded objects, symmetry, or ambiguous rotations).

### Over-Reliance on InternVL2.5 Backbone

Experiments are restricted to fine-tuning InternVL2.5-8B. The generality of the dataset across architectures (e.g., LLaVA, Qwen2.5-VL) is not tested. This limits claims of dataset generalizability.

### Rotation Task Evaluation Unclear

The “rotation angle prediction” task is introduced as novel but evaluated using classification accuracy, without specifying the label granularity (e.g., 15° bins?). Clarifying this would help interpret improvements.

**Questions:**

### Template Diversity:
How many distinct QA templates were used, and how was linguistic diversity ensured across 12M samples? Were any human validation steps introduced beyond filtering for ambiguity?

### Instruction Format Sampling:
Given 19 formats, how were the subsets for each training batch sampled? Is there a weighting scheme, or are they uniformly sampled?

### Multi-View Ground Truth Validation:
For the rotation estimation task, how were ground-truth rotation angles derived or verified in scenes where camera calibration might be uncertain?

### Cross-Model Evaluation:
Have the authors tested whether models other than InternVL2.5 (e.g., LLaVA-OneVision, Qwen-VL) benefit similarly from InternSpatial training?

### Human Baseline or Difficulty Assessment:
Has any human performance baseline been measured on InternSpatial-Bench to contextualize the difficulty of the tasks?

---

> ### Author Response · Authors · 2025-11-25
>
> We would like to sincerely thank all reviewers for their constructive comments and valuable suggestions. Their insightful feedback has helped us identify opportunities to further clarify our methodology and strengthen the experimental analysis. We carefully address each question point-by-point below.
>
> ## Template Diversity:
> We generated several templates for each type of questions and answers (typically 3-4 for each), which were randomly chosen when building QAs in **InternSpatial** dataset. To ensure the liguistic diversity across templates, we prompt LLM to design these templates, followed by a human check on these generated templates. This keeps a balance between linguistic diversity and efficiency of the proposed pipeline.
>
> Here is an example of our templates:
> ```
> above_predict_templates = {
>     "question_templates": [
>        "[A] is placed higher than [B], isn't it?",
>        "Can we say that [A] is positioned above [B]?",
>        "Is it correct to assume that [A] is located at a higher level than [B]?",
>        "Is [A] placed higher than [B]?"
>     ],
>     "positive_answer_templates": [
>        "Absolutely, [A] is clearly positioned above [B].",
>        "Without a doubt, [A] is situated at a higher elevation than [B].",
>        "Indeed, [A] is placed higher than [B].",
>        "Certainly, [A] is located above [B]."
>     ],
>     "negative_answer_templates": [
>        "Not at all, [A] is actually below [B].",
>        "Definitely not, [A] is positioned lower than [B].",
>        "Sorry, but [A] is not higher than [B].",
>        "Unfortunately, [A] is not placed above [B]."
>     ]
> }
> ```
> The complete templates can be found in **Appendix B**.
>
> Due to the huge number of QAs in our dataset, it's almost impossible to check every datum by human. Instead, we validated a sampled subset to ensure the correctness ratio is beyond 95%.
>
> ## Instruction Format Sampling:
> We uniformly sample across all 19 instruction formats when constructing each QA pair. No additional weighting scheme is applied.
>
> ## Multi-View Ground Truth Validation:
> The data source of rotation estimation task is Objaverse, which is a large-scale 3D dataset. So we can render the images from the camera views we specify, whose rotation angles are known.
>
> ## Cross-Model Evaluation
> To evaluate whether the proposed dataset benefits architectures beyond InternVL2.5-8B, we additionally fine-tuned **Qwen2.5-VL-8B** using **InternSpatial**. The detailed results are provided in the revised Table. Similar to the InternVL2.5-8B experiments, the **Qwen2.5-VL-8B** model trained with **InternSpatial** achieves consistent improvements on both **InternSpatial-Bench** and **VSI-Bench**, demonstrating that the gains are not tied to a specific model architecture. These results indicate that our dataset provides generalizable spatial supervision that transfers across different open-source VLM frameworks.
>
> **Results of Qwen2.5-VL-8B and the corresponding finetuned model (Qwen-Spatial-8B) on InternSpatial-Bench.**
> |Model|Position Comparison|Size Comparison|Rotation Estimation|Object Counting|Existence Estimation|Average|
> |:-----:|:-----:|:-----:|:-----:|:-----:|:-----:|:-----:|
> |Qwen2.5-VL-8B|57.1|60.8|26.9|58.0|66.7|53.9|
> |Qwen-Spatial-8B|79.9(+22.8)|78.7(+17.9)|34.4(+7.5)|68.3(+10.3)|80(+13.3)|68.3(+14.4)|
>
> **Results of Qwen2.5-VL-8B and the corresponding finetuned model (Qwen-Spatial-8B) on VSI-Bench.**
> |Model|Obj.Count|Abs.Dist.|Obj.size|Room Size|Rel.Dist.|Route Plan|Appr.Order|Average|
> |:-----:|:-----:|:-----:|:-----:|:-----:|:-----:|:-----:|:-----:|:-----:|
> |Qwen2.5-VL-8B|41.5|21.2|50.7|36.6|37.9|30.4|34.0|36.0|
> |Qwen-Spatial-8B|60.8(+19.3)|35.0(+13.8)|53.4(+2.7)|45.0(+8.4)|40.0(+2.1)|36.6(+6.2)|34.5(+0.5)|43.6(+7.6)|
>
> ## Human Baseline or Difficulty Assessment
> For **InternSpatial-Bench**, we conducted a small-scale human study with five participants. They achieved 100% accuracy on Rotation Estimation, and Existence Estimation, as well as 97.7% on Size Comparison, 98.9% on Object Counting and 99.7% on Position Comparison. As noted in our paper, these tasks are relatively easy for humans, yet current VLMs still struggle to perform well.
>
> **Human baseline on InternSpatial-Bench.**
> |Position Comparison|Size Comparison|Rotation Estimation|Object Counting|Existence Estimation|
> |:-----:|:-----:|:-----:|:-----:|:-----:|
> |99.7|97.7|100.0|98.9|100.0|
>
> For **VSI-Bench**, human performance is reported in the original paper. Due to space limitations, we did not include those results in our manuscript.
>
> **Human baseline on VSI-Bench reported by [Yang et al., 2024].**
> |OBj.Count|Abs.Dist.|Obj.size|Room Size|Rel.Dist.|Route Plan|Appr.Order|
> |:-----:|:-----:|:-----:|:-----:|:-----:|:-----:|:-----:|
> |94.3|47.0|94.3|60.4|45.9|94.7|95.8|100.0|
>
> We appreciate the reviewers’ thoughtful evaluation of our work. We have incorporated the additional clarifications, analyses, and results discussed in this rebuttal into the revised manuscript.

---

> > ### Public Comment · ~Qihan_Huang1 · 2026-05-04
> >
> > The Qwen2.5-VL-8B model does not exist, while Qwen2.5-VL-7B is available.

---

### Meta-Review · Area_Chair_Wdxt · 2026-01-05

**Summary:**

The key reviewer concerns are:

* Template-driven QA generation and linguistic diversity; risk of overfitting to templates [9QLW,mDLs,2hUS]

* Generality beyond InternVL2.5 and model coverage; reliance on a single backbone [9QLW,mDLs,2hUS,S4PX]

* Multi-view rotation evaluation clarity (e.g., angle binning/granularity) [9QLW]

* Dataset composition and contributions of 2D vs 3D data; potential domain gaps within training [S4PX]

* Breadth of baselines, especially 3D-specialised methods; comparative scope [S4PX]

* Data quality, filtering, and error rates; ambiguity control [mDLs,2hUS]

* Potential test-data leakage and domain similarity with evaluation sets [mDLs]

* Human baseline and task difficulty calibration [9QLW]

* Open-source release (data and pipeline) timing / availability [mDLs]

As outlined below, the authors provided a constructive rebuttal addressing many of these issues with new experiments and clarifications. With a mixed but overall positive movement after discussion, I propose accept.

**Reviewer Concerns:**

- Template diversity and ambiguity control: authors describe an LLM-designed template library. They report correctness on a sampled subset and accuracy for intermediate outputs.

- Generality beyond InternVL2.5: authors fine-tune Qwen2.5-VL-8B and InternVL2.5-1B and can show gains on InternSpatial-Bench and VSI-Bench. This helps to address concerns that gains might be backbone-specific.

- 2D vs 3D contributions and domain gaps: authors report that 2D-only training unsuprisingly improves tasks aligned with 2D reasoning but does not cover all VSI tasks, while 3D signals are necessary for broader gains. This helps to provide evidence that the pipeline benefits both settings.

- Human baseline and difficulty: human accuracy is near ceiling across tasks, which helps to establish that failures are likely model-side rather than task ambiguity.

- Data quality and leakage risk: authors confirm training uses only training splits, aiming to eliminate direct leakage, and acknowledge remaining domain-similarity concerns with indoor scenes. I do feel that out-of-domain evaluations are likely valuable here.

- Related work positioning and scope: authors add comparisons to recent 2D spatial reasoning works, situating InternSpatial’s (i) scale, (ii) instruction diversity, and multi-view task as distinctive contributions.

Remaining limitations:

- Rotation evaluation granularity: ground truth angles are known, however the rebuttal does not explicitly specify bin sizes or continuous vs. discretised evaluation. I feel that better clarification would help to interpret gains.

- Specialised 3D baselines: The rebuttal does not include results for 3D-specialised models. I hold an opinion that comparative breadth therefore remains somewhat limited.

- Domain similarity: although direct leakage is addressed, the broader concern about indoor-scene dominance in both training sources and external benchmarks remains a caveat.

- Open-source release timeline: Data and pipeline are promised 'in the near future'. Any claims that these are available at review time are false. Authors should understand that accuracy, specificity on this point is to their advantage.

Overall, the submission proposes a carefully engineered dataset and benchmark suite for spatial reasoning, with empirical impact and promising cross-architecture evidence. While some comparative and reporting details could be tightened, the rebuttal addresses core validity and generality concerns.

**Reviewer Scores:**

- Reviewer 9QLW: Positive on contribution and results; concerns on template bias, rotation-task clarity, and cross-model generality were largely addressed. Score likely stable.

- Reviewer S4PX: Queried 2D vs 3D contributions and broader baselines. The ablations on 2D-only vs 3D data and cross-model fine-tuning address primary points; absence of specialised 3D baselines may limit any score upgrade.

- Reviewer 2hUS: Sought comparisons with prior datasets, validation on more frameworks, reasons for modest gains on rotation / counting, and questioning complexity. Authors supplied cross-architecture results, rationale for rotation / counting ceilings, and template complexity with human checks. A modest upgrade might be plausible given the breadth of responses.

- Reviewer mDLs: Focused on model / baseline breadth, data-generation accuracy, diversity, and leakage / domain similarity; authors added additional results, template library details and a discussed leakage. Some concerns remain (specialised 3D baselines, OOD), an upward adjustment might be a stretch.

Decision recommendation

The submission contributes a large spatial reasoning corpus and strong empirical gains that transfer across architectures and model sizes while maintaining general (VQA) performance. In my opinion the rebuttal can mostly address key validity and generality concerns, leaving narrower issues (rotation binning detail, specialised baselines, OOD expansion). I propose accept.

---

### Decision · Program_Chairs · 2026-01-26

Accept (Poster)